# Uncovering Locally-Persistent Bias via Ceteris Paribus Fairness

## Abstract

Fairness is a key concern in Machine Learning, requiring careful consideration of how models treat individuals from different demographic groups. In this paper, we propose Ceteris Paribus Persistent-Bias-Aware (PBA) fairness and an approach to formally quantify it. PBA fairness captures the relative neural network's confidence between an input and its counterfactual (where the sensitive attribute of the original input is flipped), while numerical features are jointly perturbed within a local neighborhood, but kept identical across both instances. As such, PBA fairness allows us to isolate the effect of the sensitive attribute, enabling formal identification of disparities that are consistently present in the model behavior within an entire neighborhood. We evaluate our proposed approach under both fairness-agnostic and fairness-aware training methods and compare it to several well-established fairness metrics on three benchmark datasets: Adult, COMPAS, and German Credit. The results demonstrate that our proposed approach identifies formally-proved disparities present in the model behavior, but overlooked by other approaches, and offers additional insights into model behavior.

## 1 Introduction

Deep Neural Networks (DNNs) have achieved significant success in various domains, including healthcare, finance, and labor analytics, due to their ability to model complex patterns and make high-accuracy predictions (LeCun et al., 2015; He et al., 2016). However, their widespread adoption raises critical concerns about fairness, as models may unintentionally discriminate against individuals based on sensitive features like race or gender (John et al., 2020). This is particularly problematic when models inadvertently propagate biases inherent in the data or introduced by training algorithms (Barocas et al., 2023; Khedr & Shoukry, 2023). Ensuring that DNNs make unbiased predictions across different demographic groups is essential, especially in high-stakes applications where unfair decisions can have significant socioeconomic consequences (Mehrabi et al., 2021; Kleinberg et al., 2018; Sheng et al., 2024).

Although fairness has been extensively studied in the field of Machine Learning (ML), many existing approaches focus on group-level metrics and often overlook subtle individual-level disparities. Group fairness notions such as Statistical Parity (SP) and Equalized Odds (EO) are widely used (Mehrabi et al., 2021), but they provide limited insight into how specific inputs are treated. On the other hand, individual fairness methods such as Fairness through Awareness (Dwork et al., 2012) rely on predefined similarity metrics, while exact verification-based approaches (Biswas & Rajan, 2023; Khedr & Shoukry, 2023) often employ techniques such as SMT solving or symbolic reasoning, which can suffer from scalability issues in high-dimensional or complex architecture.

To address these limitations, we propose *Ceteris Paribus Persistent-Bias-Aware (PBA) Fairness*, a novel approach that quantifies model fairness based on relative prediction confidence between an input and its counterfactual (where a sensitive attribute of the original input is flipped), while numerical features are jointly perturbed within a local neighborhood but kept identical across both instances. As such, PBA fairness maintains "other things [features] equal", hence the term *Ceteris Paribus* (Xie et al., 2023), and allows us to isolate the marginal effect of the sensitive attribute in an entire region, enabling detection of disparities consistently present in the local neighborhood of an input. Unlike classical fairness metrics that evaluate disparities solely at the label level, PBA fairness captures the changes in model confidence within the neighborhood of an input that might

not necessarily affect the final decision. This makes it particularly effective for detecting fairness violations in settings where confidence calibration and individual-level reliability are critical, such as high-stakes decision-making and fairness-aware model training.

We conduct extensive experiments on different datasets. Among them, in the German Credit dataset, we assess whether age could influence decisions' confidences. We could establish on a fairness-aware model, that in $90\%$ of the correctly classified cases for Junior candidates, decisions were made with higher confidence just because the cases involved Junior candidates (i.e., persistently higher confidence compared to Senior counterfactuals where all other values were kept equal). This proportion drops to $5\%$ when considering correctly classified cases for Senior candidates. Here, changing a Senior case to a Junior one (while maintaining all other values to be equal) boosts confidence for more than $94\%$ of the cases. This overwhelming and persistent confidence dependency on entire neighborhoods around the considered points could only be uncovered with PBA fairness. Existing metrics, such as Statistical Parity (Dwork et al., 2012), Counterfactual Fairness Accuracy (Kusner et al., 2017), Equalized Odds (Hardt et al., 2016), or Robustness Bias (Nanda et al., 2021) (described in Section 3.3) do not account for confidence discrepancy in the treatment of Senior and Junior candidates.

Our main contributions are summarized in the following:

- We introduce PBA fairness to quantify individual-level fairness by measuring the relative change in neural network's confidence between an input and its counterfactual, where the numerical features are jointly perturbed within a local neighborhood.

- Grounded in formal methods, our approach offers verifiable guarantees and supports detection of biases persistently present in the model behavior within an input's neighborhood.

- We conduct extensive experiments on three benchmark datasets, Adult, COMPAS, and German Credit, demonstrating that PBA fairness captures formally-proved consistent disparities overlooked by classical metrics and offers additional insights into model behavior.

## 2 CETERIS PARIBUS PERSISTENT-BIAS-AWARE FAIRNESS

In this section, we propose a formulation for capturing PBA fairness by analyzing the model's behavior within a neighborhood, identifying persistent-bias relative to an input and its counterfactual. The counterfactual is generated by altering a sensitive attribute. Explicitly, we investigate the relative output confidence of a neural network on an input and its corresponding counterfactual, where both inputs share identical categorical features and are jointly perturbed over the same region of the numerical input space, differing only in the value of the sensitive attribute.

Assume a neural network, denoted by $\mathcal{N}$, with $N + 1$ layers, which maps a set of input features to a corresponding target output. Let $f : \mathbb{R}^{n_0} \to \mathbb{R}^{n_N}$ denote the function implemented by the network $\mathcal{N}$, which maps an input $\boldsymbol{x}^{(0)} \in \mathbb{R}^{n_0}$ to the output at the final layer $\boldsymbol{x}^{(N)} \in \mathbb{R}^{n_N}$. The input $\boldsymbol{x}^{(0)}$ is partitioned into sensitive attributes $\boldsymbol{s} \in \mathbb{R}^{n_s}$ and non-sensitive attributes $\boldsymbol{z} \in \mathbb{R}^{n_z}$, such that $\boldsymbol{x}^{(0)} = [\boldsymbol{s}, \boldsymbol{z}]$ and $n_0 = n_s + n_z$. The output $\boldsymbol{x}^{(N)}$ is then used to produce a predicted label $\hat{y}$, for example via a softmax or sigmoid activation depending on the task, where the true label is $y$. For each class $c_i$ in the last layer $N + 1$, the output value is $\boldsymbol{x}_{c_i}^{(N+1)} = \sigma_{c_i}(\boldsymbol{x}^{(N)})$, where $\sigma(\cdot)$ captures the softmax or sigmoid function.

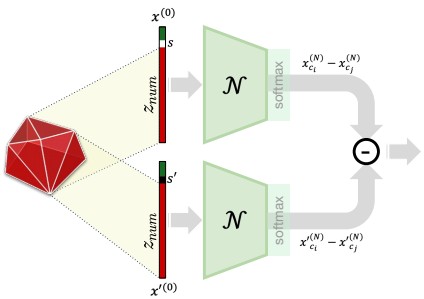

Figure 1: Overview of PBA fairness.

To implement single-attribute interventions, we construct counterfactual inputs by altering only one sensitive attribute at a time. For a given input $\boldsymbol{x}^{(0)} = [\boldsymbol{s}, \boldsymbol{z}]$, let $\boldsymbol{s}'$ denote a counterfactual version of $\boldsymbol{s}$ in which a single sensitive attribute is flipped while all other components remain unchanged, so that the corresponding counterfactual input is given by $\boldsymbol{x}'^{(0)} = [\boldsymbol{s}', \boldsymbol{z}]$.

We introduce PBA fairness to analyze the model's behavior under single-attribute interventions. PBA fairness compares the relative output confidence vectors $\boldsymbol{x}^{(N)}$ and $\boldsymbol{x}'^{(N)}$ to capture subtle changes in the model's behavior that may not alter the predicted class but still indicate fairness violations. Figure 1 provides an overview of the proposed PBA fairness. The red polytope illustrates the numerical features $\boldsymbol{z}_{num}$ that can be perturbed within a neighborhood around the original input values, where the perturbation is exactly the same for the original input and its counterfactual. To evaluate the effect of the perturbations on numerical attributes and the change in the sensitive attribute from $\boldsymbol{s}$ (sensitive attribute for the original input) to $\boldsymbol{s}'$ (flipped sensitive attribute for the counterfactual), network $\mathcal{N}$ is duplicated, and both the original $\boldsymbol{x}^{(0)}$ and counterfactual inputs $\boldsymbol{x}'^{(0)}$ are propagated through it. The difference between their logits, computed just before the softmax layer, is then analyzed to evaluate fairness. To this end, we formulate the following optimization problem, where the relative local bias of the input with respect to its counterfactual is investigated:

$$\min_{\boldsymbol{x}^{(0)}} \ (\boldsymbol{x}_{c_i}^{(N)} - \boldsymbol{x}_{c_j}^{(N)}) - (\boldsymbol{x}'^{(N)}_{c_i} - \boldsymbol{x}'^{(N)}_{c_j}), \tag{1}$$

$$\text{s.t.} \quad \tilde{\boldsymbol{x}}^{(0)} = [\boldsymbol{s}, \tilde{\boldsymbol{z}}], \quad \tilde{\boldsymbol{x}}^{(0)} \in \boldsymbol{D}, \tag{2}$$

$$\tilde{\boldsymbol{x}}'^{(0)} = [\boldsymbol{s}', \tilde{\boldsymbol{z}}], \quad \hat{y} = \hat{y}' = c_i, \tag{3}$$

$$\|\boldsymbol{z}_{\text{num}} - \tilde{\boldsymbol{z}}_{\text{num}}\|_\infty \leq \delta, \quad \tilde{\boldsymbol{z}} = [\tilde{\boldsymbol{z}}_{\text{num}}, \tilde{\boldsymbol{z}}_{\text{cat}}] \tag{4}$$

$$\boldsymbol{z} = [\boldsymbol{z}_{\text{num}}, \boldsymbol{z}_{\text{cat}}], \quad \boldsymbol{z}_{\text{cat}} = \tilde{\boldsymbol{z}}_{\text{cat}} \tag{5}$$

$$\boldsymbol{x}^{(0)} = [\boldsymbol{s}, \boldsymbol{z}], \quad \boldsymbol{x}'^{(0)} = [\boldsymbol{s}', \boldsymbol{z}] \tag{6}$$

$$\boldsymbol{x}^{(k)} = f_k(\boldsymbol{x}^{(k-1)}), \ \forall k \in \{1, \dots, N\}, \tag{7}$$

$$\boldsymbol{x}'^{(k)} = f_k(\boldsymbol{x}'^{(k-1)}), \ \forall k \in \{1, \dots, N\}. \tag{8}$$

The objective function in Equation (1) corresponds to the logarithm of the ratio of softmax outputs for classes $c_i$ and $c_j$ between the input and its counterfactual:

$$\log\left(\frac{\frac{\sigma_{c_i}(\boldsymbol{x}^{(N)})}{\sigma_{c_j}(\boldsymbol{x}^{(N)})}}{\frac{\sigma_{c_i}(\boldsymbol{x}'^{(N)})}{\sigma_{c_j}(\boldsymbol{x}'^{(N)})}}\right) = (\boldsymbol{x}_{c_i}^{(N)} - \boldsymbol{x}_{c_j}^{(N)}) - (\boldsymbol{x}'^{(N)}_{c_i} - \boldsymbol{x}'^{(N)}_{c_j}),$$

where $\sigma_{c_i}(\boldsymbol{x}^{(N)})$ captures the softmax output of class $c_i$, given the input $\boldsymbol{x}^{(N)}$ (See Proof 1 in the Appendix for details). Note that $\boldsymbol{x}_{c_i}^{(N)} - \boldsymbol{x}_{c_j}^{(N)}$ and $\boldsymbol{x}'^{(N)}_{c_i} - \boldsymbol{x}'^{(N)}_{c_j}$ represent the difference between the logit values associated to classes $c_i$ and $c_j$ for the input and its corresponding counterfactual, respectively. As such, Equation (1) introduces the objective function used to capture the logarithm of the minimum confidence difference between an input $\boldsymbol{x}^{(0)}$ and its counterfactual $\boldsymbol{x}'^{(0)}$ (for choosing $c_i$ over $c_j$) in the input region $\boldsymbol{D}_{\tilde{\boldsymbol{x}}^{(0)}}^\delta$, which includes all $\delta$-perturbed inputs in the neighborhood of the input $\tilde{\boldsymbol{x}}^{(0)}$ in the considered dataset $\boldsymbol{D}$.

The optimization is subject to the constraints defined in Equations (2)–(8). Equation (2) specifies an input from the dataset $\boldsymbol{D}$, composed of sensitive and non-sensitive attributes, denoted by $\boldsymbol{s}$ and $\tilde{\boldsymbol{z}}$, respectively. The corresponding counterfactual $\tilde{\boldsymbol{x}}'^{(0)}$ is defined in Equation (3), where one of the sensitive attributes is flipped, represented as $\boldsymbol{s}'$. Equation (3) further expresses our focus on cases where the predicted outcome remains the same, requiring $\hat{y} = \hat{y}'$, which corresponds to the true label $c_i$. Counterfactuals that lead to class changes ($\hat{y} \neq \hat{y}'$) are not included here and are addressed by separate metrics. Equation (4) enforces that the perturbed numerical attributes $\boldsymbol{z}_{\text{num}}$ lie within the $\delta$-neighborhood of $\tilde{\boldsymbol{z}}_{\text{num}}$, which is the shared non-sensitive numerical component of both the original input and its counterfactual. The non-sensitive attributes $\tilde{\boldsymbol{z}}$ also include categorical features, denoted as $\tilde{\boldsymbol{z}}_{\text{cat}}$, which are excluded from perturbation and remain fixed across all inputs within this region. In Equation (5), $\boldsymbol{z}$ is generated using $\boldsymbol{z}_{\text{num}}$ and $\boldsymbol{z}_{\text{cat}}$, where $\boldsymbol{z}_{\text{cat}}$ is equal to $\tilde{\boldsymbol{z}}_{\text{cat}}$. Equation (6) shows how the perturbed input $\boldsymbol{x}^{(0)}$ and its corresponding counterfactual $\boldsymbol{x}'^{(0)}$ are created. Equations (7) and (8) define the neurons' values across the first $N$ layers of the network for the input and its counterfactual, respectively, by expressing the output of the $k^{\text{th}}$ layer, for $k \in \{1, \dots, N\}$, as a function of the previous layer's output, using the nonlinear mapping $f_k : \mathbb{R}^{n_{k-1}} \to \mathbb{R}^{n_k}$. These equations include activation functions, introducing nonlinear constraints that make exact global optimization intractable. To overcome this, we adopt a sound over-approximation method by applying existing linear relaxations for the Rectified Linear Unit (ReLU) activation function (Ehlers,

2017; Singh et al., 2019; Baninajjar et al., 2023), which approximates the values computed at each layer using linear inequalities (See Section A.2).

## 3 EVALUATION

We evaluate our proposed approach for fairness evaluation across multiple datasets and neural network architectures. All experiments are conducted on a MacBook Pro with an 8-core CPU and 32 GB of RAM, using the Gurobi solver (Gurobi Optimization, LLC, 2023).

### 3.1 DATASETS

We evaluate PBA fairness on three widely used fairness benchmark datasets: Adult (Dua & Graff, 2019), COMPAS (Larson et al., 2016), and German Credit (Hofmann, 1994). All datasets are pre-processed prior to model training by removing rows or columns with missing values, renaming variables that reflect sensitive attributes (e.g., changing 'husband' and 'wife' to 'spouse' in the Adult dataset), normalizing features to the $[0, 1]$ range, and using one-hot encoding categorical variables. These steps ensure consistent data formatting and eliminate potential biases in variable naming.

### 3.2 NEURAL NETWORKS

Inspired by Padala & Gujar (2020), we implement four-layer fully-connected feed-forward neural networks with hidden layer configurations of $(100, 50)$, $(200, 100)$, and $(500, 100)$ neurons. The input layer size varies depending on the dataset's feature dimension, while the output layer has two neurons corresponding to the binary classification task. We adopt an $80/20$ data split, using $80\%$ for training and $20\%$ for testing, and evaluate three training methods known as fairness-agnostic, CertiFair (Khedr & Shoukry, 2023), and Fair-N (Sharma et al., 2021), with details in Section A.3.

### 3.3 FAIRNESS PROPERTIES

We define the fairness property using constraint on input perturbations for fairness analysis. Specifically, we use perturbation bounds ($\delta$) with values 1, 2, or 3, where the magnitude of $\delta$ depends on the scale of the corresponding numerical attribute. These bounds determine the size of the local neighborhood over which fairness is evaluated. For instance, in the Adult dataset, the capital gain attribute spans a large range of \$20,000, and a difference of 100 is treated as one unit of the similarity bound. For categorical attributes, we assign a similarity bound of zero, except for the sensitive attributes themselves. In our analysis, the sensitive attributes vary by dataset as we consider gender and race for the Adult dataset, gender for the COMPAS dataset, and age for the German Credit dataset.

### 3.4 FAIRNESS METRICS

We measure fairness using established metrics to evaluate demographic disparities and benchmark our approach against existing methods, with brief descriptions here and full details in Section A.4.

**Statistical Parity (SP)** requires that a predictor's positive predictions be independent of the sensitive attribute, meaning the rate of positive outcomes is unaffected by it. In this paper, we compute and report group-wise positive prediction rates, without enforcing equality (Dwork et al., 2012).

**Equalized Odds (EO)** is a group fairness criterion requiring a predictor to have equal True Positive Rate (TPR) and False Positive Rate (FPR) across groups defined by a sensitive attribute. In this paper, we report group-wise TPR, reflecting our focus on correctly classified instances, without enforcing equality (Hardt et al., 2016).

**Counterfactual Fairness Accuracy (CFA)** measures the proportion of individuals whose predicted label is unaffected by a counterfactual change in their sensitive attribute (Kusner et al., 2017).

**Robustness Bias (RB)** is a fairness metric capturing disparities in model vulnerability to adversarial perturbations across demographic groups (Nanda et al., 2021). Unlike attack-dependent empir-

ical adversarial accuracy, we quantify RB using formal verification methods that provide provable robustness guarantees, enabling reliable group-wise assessment where differences in robustness indicate potential unfairness in the model's decision boundary.

### 3.5 INTERPRETATION OF THE PBA FAIRNESS

The PBA fairness is designed to capture individual-level disparities in model confidence by comparing each input to its corresponding counterfactual. Specifically, for every input $x^{(0)}$ belonging to a subgroup, we consider a counterfactual input $x'^{(0)}$ that differs only in the sensitive attribute. Equation (1) measures the difference in output confidence between the original input and its counterfactual within a perturbation bound $\delta$. A positive value indicates that the model is more confident in its prediction for the original input than for its counterfactual counterpart within this bound.

In this paper, we aggregate individual comparisons by calculating the proportion of inputs within each subgroup that exhibit a positive objective value, meaning they exhibit higher output confidence than their counterfactual counterparts. For example, a PBA fairness score of 30% for the "Female" subgroup indicates that 30% of Female inputs demonstrate higher model confidence relative to their corresponding counterfactuals, which are generated by swapping the gender attribute from Female to Male. We refer to this setting as "observed," and define another setting as "flipped," where we swap the sensitive attribute and examine the output confidence of the flipped input with respect to its counterfactual counterpart, which is the original input. Comparing output confidence between observed and flipped settings allows evaluation of biases and model sensitivity to changes in the sensitive attribute. However, due to the inherent conservatism of the over-approximation-based technique, PBA fairness may not always be conclusively verified. As the perturbation bound increases, the relaxation becomes looser, which can slightly widen the gap between the true and estimated objective values. In such cases, verification failures may occur, not because the property is violated but because the method preserves soundness by refusing to certify properties it cannot prove with certainty.

### 3.6 RESULTS AND ANALYSIS

We evaluate the effectiveness of the proposed PBA fairness in capturing disparities in neural networks by comparing confidence scores between an input and its counterfactual differing only in sensitive features.

#### 3.6.1 ADULT DATASET

The Adult dataset contains demographic and employment-related features, and is used to predict whether an individual's income exceeds $50K per year (Dua & Graff, 2019). Table 1 shows the results of different fairness metrics, including our proposed approach, PBA, evaluated on the Adult dataset with respect to sensitive attributes, i.e., gender and race.

The upper section presents results for gender, considering the Male and Female groups. Here, standard output-based metrics such as SP, EO, and CFA, as well as RB, reveal only minor differences between the two groups. For example, in the CertiFair model with hidden layer sizes $(100, 50)$, a light gray shading appears under SP, indicating a small disparity between Male and Female outcomes, though the difference is not substantial. However, PBA fairness provides a contrasting perspective. The corresponding cells in the table show darker gray shading, indicating that PBA fairness detects a stronger disparity between Male and Female individuals, even when other metrics suggest near parity. This underscores the ability of PBA fairness to reveal more subtle, confidence-based biases that may not be captured by classical fairness metrics. Considering a perturbation bound $\delta = 1$, CertiFair exhibits the smallest disparity between Male and Female subgroups for the $(100, 50)$ network in the "observed" setting, with PBA fairness scores of 32.6% for Males and 47.5% for Females, indicating higher output confidence for Female cases relative to Males. A similar pattern is observed for the $(500, 100)$ network, where PBA fairness scores are 39.6% for Males and 50.5% for Females. In contrast, for the $(200, 100)$ network, Fair-N outperforms the other two models, exhibiting PBA fairness scores of 49.2% for Males and 37.2% for Females, resulting in the smallest disparity among the three considered models.

To further examine fairness, we compute both RB and PBA fairness under increasing perturbation bounds, enabling assessment of how these metrics respond to greater input variation. As expected,

Table 1: Fairness metrics comparison on the Adult dataset (Dua & Graff, 2019) for gender (upper) and race (lower) sensitive attributes across models trained with different methods (Khedr & Shoukry, 2023; Sharma et al., 2021) and sizes (Padala & Gujar, 2020). Network sizes are denoted as tuples, e.g., (200, 100), indicating the number of neurons in each hidden layer.

| | | | Network Size (100, 50) | | | | | | Network Size (200, 100) | | | | | | Network Size (500, 100) | | | | | |
| | | | Fairness-agnostic | | CertiFair | | Fair-N | | Fairness-agnostic | | CertiFair | | Fair-N | | Fairness-agnostic | | CertiFair | | Fair-N | |
| | pert. ($\delta$) | | Female | Male | Female | Male | Female | Male | Female | Male | Female | Male | Female | Male | Female | Male | Female | Male | Female | Male |
|---|---|---|---|---|---|---|---|---|---|---|---|---|---|---|---|---|---|---|---|---|
| SP | - | - | 51.0% | 46.8% | 39.4% | 54.0% | 50.0% | 49.8% | 51.0% | 52.0% | 42.7% | 55.3% | 44.0% | 44.8% | 47.7% | 43.5% | 42.7% | 55.0% | 49.0% | 47.7% |
| EO | TPR | - | 74.8% | 67.2% | 64.3% | 73.2% | 69.9% | 66.9% | 75.5% | 71.8% | 67.8% | 74.8% | 67.1% | 62.7% | 72.7% | 63.7% | 68.5% | 74.6% | 71.3% | 65.8% |
| CFA | - | - | 86.4% | 90.4% | 87.7% | 92.1% | 98.7% | 98.3% | 85.1% | 90.9% | 89.7% | 92.9% | 97.7% | 97.0% | 88.1% | 90.3% | 90.4% | 93.0% | 98.7% | 98.2% |
| RB observed | | 1 | 96.0% | 95.6% | 95.6% | 96.5% | 99.5% | 97.2% | 98.0% | 96.4% | 96.1% | 97.4% | 98.1% | 98.3% | 97.1% | 97.1% | 94.8% | 96.6% | 95.8% | 96.6% |
| | | 2 | 90.5% | 90.7% | 93.1% | 91.6% | 93.7% | 91.7% | 91.4% | 89.3% | 93.2% | 92.3% | 97.2% | 95.3% | 89.8% | 92.4% | 90.5% | 91.8% | 92.9% | 91.0% |
| | | 3 | 86.0% | 84.9% | 85.8% | 86.9% | 87.4% | 87.4% | 82.8% | 82.5% | 86.0% | 87.1% | 94.9% | 91.3% | 82.9% | 88.1% | 87.1% | 85.1% | 88.7% | 87.4% |
| RB flipped | | 1 | 95.7% | 97.0% | 96.9% | 95.1% | 97.4% | 98.1% | 95.8% | 95.5% | 95.7% | 95.7% | 97.0% | 99.5% | 95.4% | 98.0% | 97.6% | 93.8% | 96.6% | 97.6% |
| | | 2 | 89.8% | 90.5% | 92.5% | 87.7% | 93.7% | 94.2% | 88.7% | 91.9% | 93.8% | 90.3% | 93.6% | 96.3% | 91.2% | 91.7% | 93.5% | 86.2% | 93.1% | 94.8% |
| | | 3 | 82.7% | 85.0% | 87.0% | 83.3% | 86.0% | 90.3% | 80.1% | 84.3% | 89.4% | 84.5% | 89.3% | 93.0% | 87.0% | 87.8% | 88.6% | 79.5% | 88.1% | 87.7% |
| PBA Fairness observed | | 1 | 29.5% | 63.2% | 47.5% | 32.6% | 59.7% | 29.2% | 37.4% | 54.7% | 53.6% | 33.8% | 37.2% | 49.2% | 31.7% | 62.0% | 50.5% | 39.6% | 20.3% | 60.1% |
| | | 2 | 26.0% | 55.9% | 43.1% | 26.0% | 46.6% | 19.3% | 27.8% | 45.5% | 48.8% | 24.9% | 32.6% | 44.2% | 28.3% | 54.1% | 44.8% | 29.1% | 7.5% | 45.1% |
| | | 3 | 19.0% | 48.1% | 37.7% | 15.5% | 36.4% | 11.3% | 20.2% | 34.1% | 42.0% | 17.1% | 27.4% | 38.2% | 18.0% | 45.9% | 41.4% | 21.2% | 3.8% | 27.3% |
| PBA Fairness flipped | | 1 | 26.7% | 55.5% | 50.3% | 41.7% | 56.0% | 23.3% | 29.4% | 51.5% | 55.3% | 32.9% | 42.0% | 52.1% | 28.8% | 61.0% | 50.2% | 37.6% | 21.6% | 61.8% |
| | | 2 | 20.5% | 49.0% | 43.0% | 33.3% | 48.7% | 12.1% | 19.9% | 46.0% | 48.7% | 26.6% | 37.2% | 47.0% | 21.0% | 51.2% | 42.1% | 31.4% | 9.2% | 51.4% |
| | | 3 | 14.0% | 43.5% | 33.8% | 22.5% | 41.0% | 6.3% | 14.3% | 34.3% | 42.3% | 21.7% | 27.2% | 39.1% | 14.2% | 46.8% | 33.0% | 21.9% | 3.1% | 35.8% |
| | | | Black | White | Black | White | Black | White | Black | White | Black | White | Black | White | Black | White | Black | White | Black | White |
| SP | - | - | 34.9% | 48.1% | 37.1% | 47.2% | 40.3% | 44.8% | 38.2% | 52.8% | 35.5% | 49.1% | 41.4% | 46.9% | 29.0% | 45.0% | 36.0% | 47.8% | 40.3% | 48.3% |
| EO | TPR | - | 58.2% | 68.5% | 57.0% | 67.3% | 57.0% | 62.0% | 60.8% | 72.8% | 64.5% | 68.5% | 59.5% | 64.6% | 50.6% | 65.3% | 55.7% | 67.8% | 57.0% | 66.5% |
| CFA | - | - | 89.2% | 89.9% | 91.4% | 93.3% | 90.3% | 90.6% | 89.2% | 88.4% | 93.0% | 93.4% | 89.2% | 93.9% | 91.4% | 88.5% | 94.6% | 94.3% | 96.8% | 95.0% |
| RB observed | | 1 | 96.6% | 96.4% | 96.6% | 95.9% | 99.1% | 99.2% | 95.8% | 96.4% | 96.7% | 97.5% | 92.0% | 97.7% | 94.4% | 96.9% | 97.6% | 96.3% | 96.6% | 99.0% |
| | | 2 | 91.7% | 91.6% | 92.4% | 91.3% | 96.5% | 97.7% | 89.2% | 89.8% | 91.7% | 93.3% | 90.2% | 94.7% | 90.3% | 92.2% | 89.6% | 91.3% | 95.0% | 96.5% |
| | | 3 | 86.7% | 85.1% | 89.1% | 86.1% | 96.5% | 95.2% | 84.2% | 83.0% | 88.3% | 88.7% | 86.6% | 91.0% | 87.1% | 87.8% | 84.8% | 85.6% | 93.3% | 93.6% |
| RB flipped | | 1 | 95.8% | 94.2% | 97.1% | 98.3% | 97.2% | 100% | 95.4% | 95.8% | 96.8% | 94.2% | 96.0% | 99.1% | 96.6% | 93.5% | 96.6% | 95.2% | 96.9% | 96.6% |
| | | 2 | 90.8% | 89.2% | 92.8% | 94.1% | 95.0% | 99.1% | 90.7% | 84.2% | 92.5% | 90.8% | 89.9% | 96.4% | 91.8% | 90.3% | 92.1% | 86.4% | 93.7% | 94.1% |
| | | 3 | 85.5% | 81.7% | 88.3% | 89.9% | 94.3% | 98.2% | 85.2% | 77.5% | 86.8% | 87.5% | 86.5% | 92.0% | 87.0% | 86.3% | 86.9% | 83.2% | 91.5% | 93.3% |
| PBA Fairness observed | | 1 | 59.2% | 33.4% | 17.6% | 56.4% | 35.4% | 34.9% | 78.3% | 22.2% | 30.0% | 49.9% | 41.1% | 33.7% | 75.8% | 23.1% | 27.2% | 50.9% | 44.5% | 42.0% |
| | | 2 | 49.2% | 23.9% | 13.4% | 44.9% | 35.4% | 28.3% | 65.8% | 17.5% | 17.5% | 39.7% | 38.4% | 30.5% | 68.5% | 17.2% | 20.8% | 39.4% | 41.2% | 38.7% |
| | | 3 | 40.8% | 15.0% | 9.2% | 29.0% | 28.3% | 21.6% | 53.3% | 12.2% | 10.8% | 24.9% | 31.2% | 26.7% | 53.2% | 11.4% | 11.2% | 28.0% | 30.3% | 33.7% |
| PBA Fairness flipped | | 1 | 53.2% | 25.0% | 30.6% | 68.1% | 49.5% | 51.3% | 69.1% | 12.5% | 34.2% | 63.3% | 57.5% | 44.6% | 69.0% | 16.1% | 37.8% | 61.6% | 53.2% | 47.9% |
| | | 2 | 46.6% | 15.8% | 22.7% | 63.9% | 44.3% | 43.4% | 58.1% | 7.5% | 25.0% | 55.8% | 55.5% | 36.6% | 62.6% | 11.3% | 30.1% | 52.0% | 47.3% | 42.9% |
| | | 3 | 32.9% | 9.2% | 16.2% | 46.2% | 35.4% | 38.9% | 41.1% | 5.0% | 14.3% | 41.7% | 45.4% | 34.8% | 50.8% | 6.5% | 20.4% | 38.4% | 37.1% | 36.1% |

the RB and PBA fairness scores decrease with increasing perturbation bounds, which may indicate that the confidence of the inputs are not persistently larger than that of their counterfactual counterparts over larger perturbations or that over-approximation in the verification process leads to inconclusive results. Additionally, we include rows labeled "flipped" for both RB and PBA fairness, where the sensitive attribute (e.g., gender) of each input is explicitly switched. For example, a Female individual in the observed setting corresponds to the same individual with gender flipped to Male, as described in Section 3.5. Notably, PBA fairness scores exhibit a similar pattern between Male and Female subgroups in both the observed and flipped settings, indicating that the model's output confidence distribution remains sensitive to gender regardless of the attribute's true assignment. Meanwhile, RB scores show consistent patterns across groups and flips, with minimal differences between Males and Females, reflecting limited sensitivity to gender in this metric.

Before discussing results for the other sensitive attribute in the Adult dataset, note that fairness-agnostic models are shared across both gender and race, as they lack fairness-specific training. In contrast, CertiFair and Fair-N models are trained separately for each attribute, so different models are used for gender and race evaluations. Given this setup, the lower section of Table 1 shows only minor differences between Black and White individuals in SP, EO, CFA, and RB, suggesting minimal group-level disparity. On the other hand, PBA fairness reveals a much stronger contrast between the two groups. In fairness-agnostic models, the difference in PBA fairness scores between Black and White individuals is substantially larger, highlighting the presence of confidence-based bias that classical metrics fail to capture. This disparity is notably reduced in models trained with fairness-aware objectives. Among fairness-aware models, Fair-N outperforms CertiFair across all network sizes, showing smaller racial group differences. The fairest setup is Fair-N with hidden layers (500, 100), where PBA fairness scores for Black and White individuals, both observed and flipped inputs, approach 50% at $\delta = 1$. This suggests most samples are successfully verified and fairness estimates are less impacted by over-approximation from model non-linearities. It is important to reiterate that the PBA fairness score for one group, computed using the observed sensitive attribute, corresponds to the PBA fairness score of the other group under attribute flipping, where the sensitive attribute is switched to the opposite value. In this setup, a high PBA fairness score (e.g., close to 100%) for one group necessarily implies a low value (e.g., near 0%) for the flipped group, and vice versa. This mutual dependence prevents both values from being high at the same time. This complementarity does not hold for the RB metric, where the observed and flipped values

Table 2: Comparison of fairness metrics on the COMPAS dataset (Larson et al., 2016) by gender considering different network sizes (Padala & Gujar, 2020) and training methods (Khedr & Shoukry, 2023; Sharma et al., 2021). Network sizes are tuples indicating neurons per layer.

| | | | Network Size (100, 50) | | | | | | Network Size (200, 100) | | | | | | Network Size (500, 100) | | | | | |
| | | | Fairness-agnostic | | CertiFair | | Fair-N | | Fairness-agnostic | | CertiFair | | Fair-N | | Fairness-agnostic | | CertiFair | | Fair-N | |
| | | pert. (δ) | Female | Male | Female | Male | Female | Male | Female | Male | Female | Male | Female | Male | Female | Male | Female | Male | Female | Male |
|---|---|---|---|---|---|---|---|---|---|---|---|---|---|---|---|---|---|---|---|---|
| SP | - | - | 66.7% | 45.2% | 69.5% | 65.4% | 88.0% | 71.5% | 72.3% | 48.9% | 70.7% | 65.9% | 56.6% | 51.9% | 81.1% | 54.4% | 70.3% | 67.2% | 70.7% | 62.3% |
| EO | TPR | - | 78.3% | 61.7% | 79.6% | 80.9% | 91.1% | 83.7% | 80.9% | 66.1% | 80.9% | 81.3% | 66.2% | 65.8% | 86.0% | 72.4% | 80.9% | 82.7% | 75.8% | 75.1% |
| CFA | - | - | 90.0% | 88.0% | 91.6% | 91.6% | 89.2% | 81.3% | 89.2% | 87.7% | 91.6% | 92.2% | 96.8% | 97.9% | 85.9% | 84.7% | 91.6% | 91.4% | 95.6% | 97.2% |
| RB (observed) | | 1 | 77.7% | 87.6% | 84.1% | 86.5% | 98.6% | 95.3% | 84.9% | 88.8% | 82.3% | 86.6% | 87.2% | 90.3% | 91.2% | 86.4% | 82.4% | 86.7% | 95.3% | 93.8% |
| | | 2 | 62.4% | 70.6% | 66.9% | 67.1% | 98.6% | 87.3% | 68.4% | 74.0% | 67.1% | 66.6% | 73.7% | 78.5% | 76.9% | 71.4% | 66.7% | 67.8% | 86.0% | 85.4% |
| | | 3 | 43.9% | 51.6% | 52.2% | 48.2% | 95.9% | 80.0% | 51.3% | 55.2% | 55.1% | 49.1% | 58.3% | 65.1% | 62.6% | 51.4% | 53.5% | 50.0% | 80.0% | 76.3% |
| RB (flipped) | | 1 | 84.9% | 78.3% | 85.6% | 87.3% | 96.9% | 94.5% | 85.8% | 84.9% | 86.2% | 86.7% | 90.0% | 84.6% | 85.2% | 83.7% | 85.4% | 84.3% | 94.2% | 92.7% |
| | | 2 | 67.6% | 63.1% | 69.2% | 72.0% | 95.7% | 84.9% | 69.3% | 69.1% | 71.7% | 70.9% | 74.6% | 75.6% | 71.9% | 67.3% | 70.6% | 69.8% | 86.1% | 86.7% |
| | | 3 | 47.2% | 40.8% | 52.9% | 51.6% | 93.9% | 80.1% | 50.2% | 53.3% | 53.8% | 51.9% | 58.9% | 62.8% | 55.2% | 49.0% | 52.1% | 54.7% | 76.8% | 78.7% |
| PBA Fairness (observed) | | 1 | 47.8% | 41.0% | 7.0% | 50.6% | 28.8% | 43.3% | 58.6% | 34.4% | 6.3% | 50.6% | 39.7% | 53.9% | 71.4% | 12.4% | 5.7% | 59.0% | 32.0% | 65.0% |
| | | 2 | 38.9% | 24.2% | 3.2% | 26.9% | 21.9% | 33.5% | 44.7% | 20.0% | 2.5% | 26.7% | 34.6% | 44.7% | 56.5% | 9.4% | 1.3% | 39.9% | 23.3% | 46.8% |
| | | 3 | 27.4% | 21.5% | 1.9% | 15.9% | 18.5% | 29.0% | 31.6% | 14.4% | 1.3% | 13.0% | 28.2% | 31.7% | 48.3% | 8.9% | 0.60% | 15.6% | 4.7% | 9.5% |
| PBA Fairness (flipped) | | 1 | 41.7% | 23.6% | 14.4% | 58.6% | 34.9% | 51.4% | 48.1% | 19.1% | 14.2% | 55.7% | 38.9% | 55.8% | 61.7% | 3.4% | 11.4% | 66.0% | 30.0% | 63.3% |
| | | 2 | 34.1% | 12.7% | 9.5% | 29.9% | 29.6% | 34.2% | 40.3% | 9.9% | 7.8% | 28.5% | 34.3% | 45.5% | 49.9% | 1.4% | 2.7% | 42.1% | 19.3% | 47.3% |
| | | 3 | 23.2% | 12.1% | 4.2% | 17.8% | 25.7% | 27.4% | 26.6% | 9.2% | 1.9% | 12.7% | 24.4% | 30.1% | 40.9% | 1.4% | 0.3% | 15.1% | 4.5% | 30.0% |

are independent. The results underscore the significance of PBA fairness in revealing disparities undetected by classical metrics, while highlighting the role of fairness-aware training.

### 3.6.2 COMPAS DATASET

The COMPAS dataset contains criminal and demographic features and predicts two-year recidivism. Table 2 shows the results for several fairness metrics, including our proposed approach, PBA fairness, as applied to the COMPAS dataset with respect to the sensitive attribute gender. The results across all metrics show relatively small differences between Male and Female groups. However, the largest disparity is observed in the PBA fairness for the fairness-agnostic model with hidden layer sizes $(500, 100)$, where at $\delta = 1$, PBA fairness score is $71.4\%$ for the Female group and $12.4\%$ for the Male group in the observed setting. A disparity of a similar magnitude is present for the flipped setting, reflecting the fairness disparity. Notably, for the fairness-agnostic model, the PBA fairness score for Female individuals is substantially higher than for Males, whereas this trend is reversed for both the CertiFair and Fair-N models.

Nonetheless, selecting a single model requires a thorough assessment that considers multiple fairness metrics simultaneously. For instance, for the Fair-N model with hidden layers $(100, 50)$, SP is $88.0\%$ for the Female subgroup and $71.5\%$ for the Male subgroup, and EO is $91.1\%$ for Females and $83.7\%$ for Males, which are higher than all other models, although not as close between subgroups as in CertiFair. Moreover, the model does not achieve the smallest PBA disparity between subgroups. In contrast, comparison of the RB for the observed and flipped settings shows higher and relatively close RB values for the $(100, 50)$ Fair-N model. In the observed setting, RB is $98.6\%$ for the Female subgroup and $95.3\%$ for the Male subgroup, while in the flipped setting, it is $96.9\%$ for Females and $94.5\%$ for Males. Conversely, CFA is higher for the $(200, 100)$ Fair-N model, where it reaches $96.8\%$ for the Female subgroup and $97.9\%$ for the Male subgroup. Taken together, these results demonstrate the need to consider multiple fairness criteria when selecting a model. Employing a variety of metrics provides a more comprehensive understanding of fairness and enables well-grounded model selection tailored to specific application needs.

### 3.6.3 GERMAN CREDIT DATASET

The German Credit dataset, with financial and personal attributes, predicts credit risk. Table 3 presents the results of various fairness metrics, including our proposed approach, PBA fairness, evaluated on the German Credit dataset with the age group as the sensitive attribute. The results indicate that Fair-N demonstrates notable performance across the output-based fairness metrics, including SP, EO, and CFA, indicating equitable predictive outcomes for both age-based subgroups. Furthermore, all model configurations achieve relatively high scores on RB, suggesting consistent subgroup representation in the model predictions, with Fair-N still achieving the best performance, particularly under large perturbation values.

The PBA fairness scores, however, reveal another important dimension. Across multiple models of varying sizes and training methods, most of individuals are successfully verified, particularly for the

Table 3: Fairness metrics on the German Credit dataset (Hofmann, 1994) using age group as the sensitive attribute for different network sizes (Padala & Gujar, 2020) and training methods (Khedr & Shoukry, 2023; Sharma et al., 2021). Network sizes are tuples indicating neurons per layer.

| | | pert. (δ) | Network Size (100, 50) | | | | | | Network Size (200, 100) | | | | | | Network Size (500, 100) | | | | | |
|---|---|---|---|---|---|---|---|---|---|---|---|---|---|---|---|---|---|---|---|---|
| | | | Fairness-agnostic | | CertiFair | | Fair-N | | Fairness-agnostic | | CertiFair | | Fair-N | | Fairness-agnostic | | CertiFair | | Fair-N | |
| | | | Junior | Senior | Junior | Senior | Junior | Senior | Junior | Senior | Junior | Senior | Junior | Senior | Junior | Senior | Junior | Senior | Junior | Senior |
| SP | - | - | 76.3% | 81.5% | 68.4% | 74.1% | 97.4% | 92.0% | 76.3% | 75.3% | 65.8% | 69.8% | 92.1% | 79.0% | 68.4% | 77.8% | 60.5% | 66.0% | 94.7% | 85.8% |
| EO TPR | - | - | 85.0% | 91.0% | 80.0% | 85.2% | 100% | 97.5% | 85.0% | 86.9% | 80.0% | 79.5% | 100% | 88.5% | 85.0% | 86.1% | 80.0% | 75.4% | 100% | 94.3% |
| CFA | - | - | 94.7% | 93.8% | 100% | 96.9% | 100% | 97.5% | 97.4% | 94.4% | 100% | 94.4% | 86.8% | 90.1% | 94.7% | 92.0% | 100% | 97.5% | 92.1% | 91.4% |
| RB observed | | 1 | 100% | 100% | 100% | 97.6% | 95.2% | 100% | 100% | 96.7% | 100% | 98.3% | 100% | 98.3% | 100% | 99.1% | 96.3% | 97.4% | 100% | 99.2% |
| | | 2 | 95.7% | 97.6% | 100% | 94.4% | 95.2% | 99.2% | 100% | 94.3% | 96.0% | 95.7% | 100% | 95.8% | 100% | 96.5% | 96.3% | 86.8% | 100% | 99.2% |
| | | 3 | 95.7% | 95.2% | 100% | 92.7% | 95.2% | 99.2% | 100% | 90.2% | 88.0% | 86.2% | 100% | 95.0% | 92.0% | 93.9% | 92.6% | 84.2% | 100% | 98.3% |
| RB flipped | | 1 | 97.6% | 100% | 99.2% | 100% | 99.2% | 100% | 96.7% | 100% | 97.4% | 100% | 99.2% | 100% | 97.4% | 100% | 95.6% | 100% | 100% | 100% |
| | | 2 | 95.2% | 95.7% | 98.4% | 100% | 98.4% | 100% | 91.1% | 100% | 93.1% | 88.0% | 96.6% | 100% | 93.0% | 100% | 86.8% | 96.3% | 99.2% | 100% |
| | | 3 | 95.2% | 95.7% | 94.4% | 100% | 98.4% | 100% | 88.6% | 100% | 88.8% | 84.0% | 96.6% | 100% | 91.3% | 92.0% | 83.3% | 85.2% | 98.3% | 100% |
| PBA Fairness observed | | 1 | 21.7% | 80.0% | 33.3% | 49.2% | 85.7% | 19.0% | 45.5% | 56.9% | 64.0% | 44.0% | 85.7% | 9.2% | 48.0% | 57.4% | 44.4% | 42.1% | 90.0% | 4.2% |
| | | 2 | 21.7% | 77.6% | 29.2% | 47.6% | 85.7% | 18.3% | 45.5% | 54.5% | 60.0% | 41.4% | 85.7% | 9.2% | 40.0% | 56.5% | 44.4% | 40.4% | 90.0% | 4.2% |
| | | 3 | 17.4% | 76.0% | 29.2% | 46.8% | 81.0% | 16.7% | 40.9% | 52.8% | 52.0% | 39.7% | 85.7% | 9.2% | 40.0% | 53.0% | 44.4% | 38.6% | 90.0% | 4.2% |
| PBA Fairness flipped | | 1 | 16.0% | 78.3% | 48.4% | 54.2% | 77.0% | 9.5% | 37.4% | 50.0% | 48.3% | 36.0% | 90.8% | 14.3% | 48.0% | 48.0% | 51.8% | 51.9% | 95.8% | 10.0% |
| | | 2 | 15.2% | 73.9% | 45.2% | 54.2% | 75.4% | 9.5% | 37.4% | 50.0% | 46.6% | 36.0% | 90.8% | 14.3% | 39.1% | 48.0% | 48.2% | 44.4% | 95.8% | 10.0% |
| | | 3 | 12.8% | 69.6% | 42.7% | 54.2% | 74.6% | 9.5% | 35.0% | 50.0% | 42.2% | 32.0% | 90.8% | 14.3% | 33.9% | 48.0% | 43.0% | 40.7% | 95.8% | 10.0% |

smallest perturbation value $\delta = 1$, reflecting both high model output confidence and minimal over-approximation. For example, for the CertiFair model with a network size of $(200, 100)$, $64.0\%$ of observed Junior individuals and $36.0\%$ of flipped Senior individuals are verified, summing to $100\%$, which means that all individuals who were originally Junior are verified. A further pattern emerges in the form of a substantial disparity between Junior and Senior individuals across all Fair-N models and in the fairness-agnostic model with architecture $(100, 50)$. This gap indicates that high overall verification coverage does not translate into equitable outcomes across subgroups. It is noteworthy that CertiFair models yield a low PBA gap between Junior and Senior individuals, suggesting more balanced verification across age groups, but slightly lower SP and EO scores than Fair-N. While this does not suggest a fundamental trade-off between PBA and output-based fairness metrics, it underscores the challenge of achieving uniformly high performance across diverse fairness dimensions.

### 3.6.4 VERIFIED OUTCOME DISTRIBUTION

This section presents a detailed analysis of the PBA fairness scores to better understand how the model behaves across demographic groups, focusing on desirable and undesirable outcomes. In each figure, the left image corresponds to the observed setting and the right image to the flipped setting. As discussed earlier, in the observed setting the true subgroup for each sensitive attribute is preserved, e.g., if an individual is White, we keep it as White. In the flipped setting, the subgroup label for the sensitive attribute is swapped. Since the dataset encodes color as either Black or White, flipping means that if an individual is shown as White, their true subgroup was Black, and vice versa. This allows us to further examine the effect of the sensitive attribute on the results.

Figure 2 presents the PBA fairness scores for the Fair-N model trained on the Adult dataset using a $(500, 100)$ architecture, showing the proportion of verified desirable (HIN: high income) and undesirable (LIN: low income) outcomes across racial groups. Figure 2 shows that whether an individual is originally identified as Black or their race is flipped to Black, the model tends to be more confident in assigning a LIN outcome compared to White individuals. Conversely, when an individual is either actually White or their race is flipped to White, there is a higher PBA fairness associated with a HIN outcome. These patterns indicate that the model's confidence is influenced not only by the individual's actual demographic characteristics, but also by the demographic information as it is presented to the model in the input, whether it reflects the true subgroup or a flipped version. Such discrepancies reveal potential biases in the model, suggesting that racial representation in the input can shape the confidence in the predicted outcomes, independent of the individual's actual attributes.

Figure 3 illustrates the PBA fairness scores for the Fair-N model trained on the German Credit dataset with a $(500, 100)$ architecture, showing the proportion of verified desirable (GCR: good credit) and undesirable (BCR: bad credit) outcomes across age groups. Figure 3 shows that there is no instance where the model exhibits higher confidence in BCR outcomes for Seniors compared to Juniors, nor any instance where Juniors receive higher confidence in GCR outcomes relative to Seniors. This indicates that the model's confidence aligns with age groups, favoring Seniors for desirable outcomes and Juniors for undesirable ones.

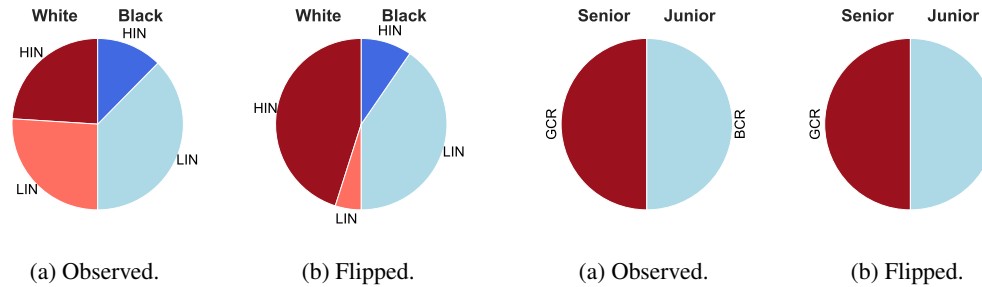

| (a) Observed. | (b) Flipped. | (a) Observed. | (b) Flipped. |

Figure 2: PBA fairness results for the Fair-N model on the Adult dataset (architecture: $(500, 100)$), showing verified desirable (HIN: high income) and undesirable (LIN: low income) outcomes for each racial group.

Figure 3: PBA fairness results for the Fair-N model on the German Credit dataset (architecture: $(500, 100)$), showing verified desirable (GCR: good credit) and undesirable (BCR: bad credit) outcomes for each age group.

**Processing Time** Average processing times and standard deviations ($\mu \pm \sigma$, in seconds) are reported across three perturbation levels and datasets. For a network of size $(100, 50)$, times increase from $0.24 \pm 0.03$ at level 1 to $0.30 \pm 0.03$ at level 2, and $0.36 \pm 0.03$ at level 3. With a larger network $(200, 100)$, they rise to $0.90 \pm 0.12$, $1.30 \pm 0.19$, and $1.66 \pm 0.21$ across the three levels. At the largest size $(500, 100)$, times further increase to $2.15 \pm 0.40$, $3.04 \pm 0.65$, and $3.82 \pm 0.69$. These results indicate that computational cost is affected by both perturbation level and network size.

## 4 RELATED WORK

Fairness in ML is typically categorized into individual and group fairness. Individual fairness requires similar individuals to receive similar outcomes (Dwork et al., 2012), while group fairness focuses on achieving parity in predictive performance across different demographic groups (Hardt et al., 2016). Common metrics such as SP, EO, and CFA (Kusner et al., 2017) have been widely used to measure fairness, but they often rely solely on predicted labels and fail to capture more subtle aspects of model behavior.

The overwhelming majority of existing fairness metrics focus on binary outcomes, overlooking disparities in model confidence that can reveal more nuanced biases (Nanda et al., 2021; Jovanović et al., 2023). While FARE (Jovanović et al., 2023) provides fairness certificates in representation space, and Robustness Bias (RB) (Nanda et al., 2021) leverage confidence scores without a counterfactual view, our method directly compares predictions of inputs and their counterfactuals within a shared, locally perturbed neighborhood. This enables a more interpretable assessment of how sensitive attributes affect model confidence.

Formal verification techniques have been applied to assess fairness in neural networks, offering guarantees under either counterfactual or distributional definitions of fairness (Wicker et al., 2023; Athavale et al., 2024; Xie et al., 2023). While these methods provide strong formal assurances, they rely on global or distributional formulations of fairness, which can limit their ability to capture local individual behavior. Our work addresses this gap by introducing a formally grounded, confidence-based approach that operates on local counterfactual comparisons, enabling assessments of how sensitive attributes influence prediction confidence.

## 5 CONCLUSION

Fairness has become a key concern in ML. In this paper, we proposed PBA fairness and an approach to formally quantify it within an entire neighborhood, enabling the identification of disparities that are consistently present in the model behavior within this neighborhood. We evaluated our proposed approach considering both fairness-agnostic and fairness-aware training methods and compared it based on the well-established Adult, COMPAS, and German Credit datasets. The results demonstrated that our proposed approach identifies formally-proved disparities and offers additional insights into model behavior.

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

## A  APPENDIX

**Disclosure of LLM usage:** Parts of this paper (e.g., TL;DR) were polished with the assistance of a large language model (ChatGPT). The authors reviewed and verified all content, and all scientific contributions, claims, and results are entirely their own.

### A.1  PROOFS FROM SECTION 2

**Lemma 1** *Let $(c_i, c_j)$ denote a pair of classes in the network $\mathcal{N}$. We consider $\boldsymbol{x}^{(0)}$ and its counterfactual $\boldsymbol{x'}^{(0)}$ as two distinct inputs. Then:*

$$\log\left(\frac{\frac{\sigma_{c_i}(\boldsymbol{x}^{(N)})}{\sigma_{c_j}(\boldsymbol{x}^{(N)})}}{\frac{\sigma_{c_i}(\boldsymbol{x'}^{(N)})}{\sigma_{c_j}(\boldsymbol{x'}^{(N)})}}\right) = (\boldsymbol{x}_{c_i}^{(N)} - \boldsymbol{x}_{c_j}^{(N)}) - (\boldsymbol{x'}_{c_i}^{(N)} - \boldsymbol{x'}_{c_j}^{(N)}).$$

**Proof 1** *By applying the* $\log$ *function on the output of the network $\mathcal{N}$:*

$$\log\left(\frac{\frac{\sigma_{c_i}(\boldsymbol{x}^{(N)})}{\sigma_{c_j}(\boldsymbol{x}^{(N)})}}{\frac{\sigma_{c_i}(\boldsymbol{x'}^{(N)})}{\sigma_{c_j}(\boldsymbol{x'}^{(N)})}}\right) = \log\left(\frac{\sigma_{c_i}(\boldsymbol{x}^{(N)}).\sigma_{c_j}(\boldsymbol{x'}^{(N)})}{\sigma_{c_j}(\boldsymbol{x}^{(N)}).\sigma_{c_i}(\boldsymbol{x'}^{(N)})}\right) =$$

$$\log\left(\frac{\frac{e^{\boldsymbol{x}_{c_i}^{(N)}}}{\sum_{u=1}^{n_N} e^{\boldsymbol{x}_u^{(N)}}} \cdot \frac{e^{\boldsymbol{x'}_{c_j}^{(N)}}}{\sum_{u=1}^{n_N} e^{\boldsymbol{x'}_u^{(N)}}}}{\frac{e^{\boldsymbol{x}_{c_j}^{(N)}}}{\sum_{u=1}^{n_N} e^{\boldsymbol{x}_u^{(N)}}} \cdot \frac{e^{\boldsymbol{x'}_{c_i}^{(N)}}}{\sum_{u=1}^{n_N} e^{\boldsymbol{x'}_u^{(N)}}}}\right) = \log\left(\frac{e^{\boldsymbol{x}_{c_i}^{(N)}} \cdot e^{\boldsymbol{x'}_{c_j}^{(N)}}}{e^{\boldsymbol{x}_{c_j}^{(N)}} \cdot e^{\boldsymbol{x'}_{c_i}^{(N)}}}\right) =$$

$$(\boldsymbol{x}_{c_i}^{(N)} + \boldsymbol{x'}_{c_j}^{(N)}) - (\boldsymbol{x}_{c_j}^{(N)} + \boldsymbol{x'}_{c_i}^{(N)}) =$$

$$(\boldsymbol{x}_{c_i}^{(N)} - \boldsymbol{x}_{c_j}^{(N)}) - (\boldsymbol{x'}_{c_i}^{(N)} - \boldsymbol{x'}_{c_j}^{(N)}).$$

### A.2  RELAXATIONS FOR EQUATIONS (7) AND (8)

We provide a sound over-approximation of Equations (7) and (8), allowing the minimization problem to be addressed via linear programming techniques. Since ReLUs are the most commonly used activation functions in neural networks, our analysis focuses on ReLU layers. However, it can be extended to any nonlinear activation function that admits a piecewise linear representation.

A ReLU activation function consists of two linear segments, forming a piecewise linear map. Consider the $i^{\text{th}}$ neuron in layer $k$, whose pre-activation value is defined as $\hat{x}_i^{(k)}$. The ReLU output is $\hat{x}_i^{(k)}$ if this value is non-negative, and zero otherwise. When inputs are perturbed within a $\delta$-radius neighborhood, each pre-activation $\hat{x}_i^{(k)}$ is bounded by lower and upper limits, $\underline{\hat{x}}_i^{(k)}$ and $\overline{\hat{x}}_i^{(k)}$, respectively. This yields three cases for the activation: always active if both bounds are non-negative, always inactive if both are negative, or uncertain if the bounds straddle zero. To handle the uncertain case in our optimization framework, we adopt the approach from (Ehlers, 2017) by approximating the ReLU with the smallest convex region bounded by these limits. This region is characterized by the inequalities:

$$x_i^{(k)} \leq \overline{\hat{x}}_i^{(k)} \cdot \frac{\hat{x}_i^{(k)} - \underline{\hat{x}}_i^{(k)}}{\overline{\hat{x}}_i^{(k)} - \underline{\hat{x}}_i^{(k)}}, \quad x_i^{(k)} \geq \hat{x}_i^{(k)}, \quad x_i^{(k)} \geq 0.$$

We compute these bounds by propagating the input bound through the network, layer by layer. Our framework supports a variety of layer types, including convolution, zero-padding, max-pooling, permutation, and flattening layers.

### A.3  MODEL TRAINING DETAILS

**Fairness-agnostic**  As a baseline, we train the model without incorporating any fairness-specific objectives. The Adam optimizer is used with a learning rate of 0.001, and sparse categorical cross-

entropy serves as the loss function. This setup allows assessment of the model trained exclusively to optimize accuracy without incorporating fairness regularization.

**CertiFair**    Following the CertiFair framework (Khedr & Shoukry, 2023), we train our model using a combined loss function that balances classification accuracy and fairness. Fairness is promoted by generating counterfactual inputs in which the sensitive attributes are swapped, and penalizing the model when its predictions differ significantly between the original and flipped instances. The total training loss is a weighted sum of the classification loss and the fairness loss, with the local fairness regularization parameter $\lambda_f$ controlling the trade-off. In our experiments, consistent with (Khedr & Shoukry, 2023), we set $\lambda_f$ to $0.95$ for Adult, $0.9$ for COMPAS, and $0.2$ for German Credit.

**Fair-N**    The Fair-N framework (Sharma et al., 2021) integrates fairness and robustness objectives into the training loss of a neural network. The total loss is a weighted sum of three components: cross-entropy classification loss, a fairness loss, and a robustness loss. Fairness is enforced by penalizing disparities in average confidence margins, i.e., the absolute difference between class logits, across different groups. Robustness is encouraged through larger margins between predicted classes, with the robustness loss defined as the inverse of the average distance to the decision boundary. In our experiments, we consider $\lambda_r = 0$ and $\lambda_f = 1$ to focus only on maximizing fairness.

## A.4    DETAILS ON FAIRNESS METRICS

**Counterfactual Fairness Accuracy (CFA)**    Formally, for a prediction function $f$, CFA is defined as:
$$\text{CFA} = \mathbb{P}\big(f(X, A = a) == f(X', A = a')\big),$$
where $X'$ denotes the counterfactual input derived by changing the sensitive attribute from $a$ to $a'$, while keeping other features fixed. CFA is derived from an individual-level fairness metric proposed by Kusner et al. (2017).

**Statistical Parity (SP)**    A predictor $\hat{Y}$ satisfies SP if
$$\mathbb{P}(\hat{Y} = 1 \mid A = a) = \mathbb{P}(\hat{Y} = 1 \mid A = a') \quad \forall a, a'.$$

This metric measures group-level fairness by checking whether the probability of a positive prediction is the same across all groups defined by the sensitive attribute (Dwork et al., 2012).

**Equalized Odds (EO)**    A predictor $\hat{Y}$ satisfies EO if
$$\mathbb{P}(\hat{Y} = 1 \mid Y = y, A = a) = \mathbb{P}(\hat{Y} = 1 \mid Y = y, A = a') \quad \forall y \in \{0, 1\}, \forall a, a'.$$

This metric measures group-level fairness by ensuring that the probability of a positive prediction is the same across groups for each true outcome, which enforces equal true positive and false positive rates (Hardt et al., 2016). Similar to SP, we report group-wise rates without enforcing equality and focus exclusively on the TPR component.

**Robustness Bias (RB)**    Let $\text{Acc}_{\text{rob}}(a)$ denote the robustness accuracy of the model on group $A = a$ under a specified perturbation $\delta$, computed via formal verification. Disparities in RB are then assessed by comparing robustness accuracy values across different groups.

