# OpenReview forum: "Uncovering Locally-Persistent Bias via Ceteris Paribus Fairness"
_ICLR.cc/2026/Conference — Submitted to ICLR 2026_

### Official Review · Reviewer_XuoK · 2025-10-15

**Soundness:** 2
**Presentation:** 3
**Contribution:** 1
**Rating:** 0
**Confidence:** 5

**Summary:**

The authors of this paper introduce 'Ceteris Paribus Persistent-Bias-Aware (PBA) Fairness', an approach for evaluating individual-level fairness in neural networks. The proposed approach compares a model’s confidence between an input and its counterfactual, while jointly perturbing numerical features within a small local neighbourhood held constant across both instances. By examining differences in logit-level confidence across these matched pairs, the authors aim to quantify persistent local bias that may not affect classification labels but could reveal latent disparities in model behaviour.

**Strengths:**

The strengths of this paper are:
- The paper is clear in terms of notation and formalisation. It is easy to follow along and read for a general audience and experts.
- It does a good job of analysing the result and interpreting what they actually mean for a reader, basically, in terms of clarity it is good.

**Weaknesses:**

The weaknesses of this paper are:
- The proposed method is highly derivative and therefore the novelty is low. The method: flips the sensitive attribute only to measure the effect of the model's output (Counterfactual Fairness- Kusner et al. 2017), the method also applies small pertubations to the data (Fairness Through Robustness- Nanda et al. 2021).This method as a whole seems like a trivial combination of multiple significant papers in the literature.
- The paper also clearly fails to preserve the strong logic of Nanda et al. 2021. If I am correct, by conditioning \hat{y} = \hat{y}' = c_i, this excludes instances where flipping the sensitive attribute changes the model’s decision: the very cases that define individual bias in both counterfactual and robustness-based formulations. As a result, the metric reduces to a confidence-difference statistic over already-consistent predictions, losing the core insight of Nanda et al.’s framework, which measures groupwise disparities in decision vulnerability. The method, thus replicates existing fairness ideas incorrectly, yielding a less meaningful method.
- Equation 9 (which is unlabelled) is a logit difference interpreted as confidence. I am worried the resulting comparisons in the evaluation. If I recall correctly, CertiFair and Fair-N use additional constraints in their training objective. Thus, the logit mangitudes will change between comparable models, making comparison unfair. To further clarify, without temperature scaling or normalisation then I can't be sure the reported metrics are correct.

**Questions:**

- Could you clarify what specific methodological or conceptual contribution is new beyond this combination? In what way does CPPBA offer capabilities that are not already achievable by existing counterfactual or robustness-based fairness frameworks?
- Can you explain the rationale for the \hat{y} = \hat{y}' = c_i restriction, given that such decision-flip cases are typically the most consequential indicators of individual unfairness?
- Equation (9) (the logit difference used as a “confidence” measure) appears to depend directly on raw logit magnitudes. Since models such as CertiFair and Fair-N are trained with additional constraints or penalties that alter the logit scale, how do you ensure that CPPBA scores remain comparable across models?

---

> ### Author Response · Authors · 2025-11-24
>
> We thank the reviewer for the careful reading and for highlighting both strengths and concerns. We respond to the points below.
>
> **Regarding the ethics flag (research integrity)**
>
> The work is original. It is not a dual submission, and it does not reuse text or results from prior publications. Prior work on counterfactual fairness and robustness bias is fully cited, and we do not claim those ideas as new. We also note that the selections “no ethics review needed” and “research integrity issues” seem inconsistent, perhaps due to a clerical error. In case this is not an error, we expect a clear justification for such claims in the review, which we don’t seem to find for now.
>
>
> > The proposed method is highly derivative and therefore the novelty is low. The method: flips the sensitive attribute only to measure the effect of the model's output (Counterfactual Fairness- Kusner et al. 2017), the method also applies small perturbations to the data (Fairness Through Robustness- Nanda et al. 2021).This method as a whole seems like a trivial combination of multiple significant papers in the literature.
>
> >Could you clarify what specific methodological or conceptual contribution is new beyond this combination? In what way does CPPBA offer capabilities that are not already achievable by existing counterfactual (Kusner et al. 2017) or robustness-based (Nanda et al. 2021)  fairness frameworks?
>
> **From an experimental point of view, the results clearly demonstrate the value of CPPBA, i.e., CPPBA outperforms Kusner et al. and Nanda et al. by a large margin. This is shown in Tables 1, 2, and 3, where CPPBA consistently detects bias instances that RB (Nanda et al.) and CFA (Kusner et al.) cannot.**
>
> In the experiments, we explicitly show settings where SP, EO, CFA, and RB all indicate minimal variation between groups, yet CPPBA reveals large confidence gaps, even for fairness-aware models. These are cases where a model looks fair in terms of decisions and robustness, but remains more confident for some groups. Counterfactual fairness would call them fair because the label stays the same, and RB would view them as robust. CPPBA shows that there is still a formally-verifiable disparity.
>
> To illustrate this, we include several examples from our experiments. Considering Table 1 on the Adult dataset (gender, $\delta = 1$), CertiFair (100,50) shows almost no disparity in RB (95.6% vs. 96.5%) and only minor gaps in SP, EO, and CFA. Yet CPPBA uncovers a much larger confidence gap: 47.5% for Females versus 32.6% for Males. Another example is in Table 2 on COMPAS, a fairness-agnostic model (500, 100) shows modest SP and EO disparity, while CPPBA reveals a wide 71.4% vs. 12.4% gap between Female and Male individuals.
>
> **From a methodological point of view, CPPBA provides formal guarantees for all the points in an entire region, something that neither Kusner et al. nor Nanda et al. could do. Kusner et al. only considers one point. Nanda et al. check robustness only in a statistical sense, with no hard formal guarantees.**
>
> **More importantly, CPPBA considers the exact same perturbation for the input and its counterfactual, for all possible perturbations in an entire region with hard formal guarantees, for the first time. Formulating this in a sound (in model checking terminology) way in formal methods is one of our main contributions.**
>
> RB evaluates each input independently and measures how easily its decision changes. CPPBA instead couples an input $x$ with its counterfactual $x'$ and applies the exact same perturbations to both. Nanda et al. analyse robustness for a single trajectory $x \mapsto f(x)$ under perturbations. In contrast, our setting verifies a relational property between two coupled trajectories $x \mapsto f(x)$ and $x' \mapsto f(x')$, using the exact same numeric perturbation $z \to \tilde{z}$. This allows us to assess, across the entire region, whether the logit margin for class $c_i$ on $x$ consistently lies above/below the corresponding margin on its counterfactual $x'$, effectively comparing two linked executions of the model under same perturbations.
>
> CPPBA considers the exact same perturbation for the input and its counterfactual, for all possible perturbations in an entire region with hard formal guarantees. We formulate this as a sound problem in formal methods and show that CPPBA consistently detects bias instances that RB (Nanda et al.) and CFA (Kusner et al.) cannot.
>
> To summarize, our main contributions are (consistent with the original contributions listed in Introduction Section of our submission):
>
> * define and formalize locally-persistent confidence-based bias between an input and its counterfactual under shared perturbations,
> * provide a sound verification procedure tailored to this relational property, and
> * show empirically (Tables 1, 2, 3) that this reveals fairness issues missed by both label-based and robustness-based metrics.

---

> > ### Author Response · Authors · 2025-11-24
> >
> > > The paper also clearly fails to preserve the strong logic of Nanda et al. 2021. If I am correct, by conditioning \hat{y} = \hat{y}' = c_i, this excludes instances where flipping the sensitive attribute changes the model’s decision: the very cases that define individual bias in both counterfactual and robustness-based formulations. As a result, the metric reduces to a confidence-difference statistic over already-consistent predictions, losing the core insight of Nanda et al.’s framework, which measures groupwise disparities in decision vulnerability. The method, thus replicates existing fairness ideas incorrectly, yielding a less meaningful method.
> >
> > >Can you explain the rationale for the \hat{y} = \hat{y}' = c_i restriction, given that such decision-flip cases are typically the most consequential indicators of individual unfairness?
> >
> > **Our proposed framework can be readily adjusted to capture the cases where flipping the sensitive attribute changes the model’s decision.**
> >
> > This condition $\hat{y} = \hat{y}' = c_i$ can be readily adjusted to allow the framework to incorporate cases in which the predicted label of the counterfactual is incorrect. However, when the prediction changes *under the counterfactual*, the optimization naturally yields an objective value below zero. This indicates that the counterfactual receives a lower margin than the original instance within the same perturbation region. In this way, flipped decisions are captured as negative-objective outcomes and do not fall outside the scope of the method. We do not need to change our framework, only consider and feed the samples where flipping the sensitive attribute changes the model’s decision. As an example, we examined the results for the COMPAS dataset using the (200, 100) model.
> >
> > |            |            | Fairness-agnostic |      | CertiFair |      | FairN  |      |
> > |------------|------------|-------------------|------|-----------|------|--------|------|
> > |            |            | Female            | Male | Female    | Male | Female | Male |
> > | $\delta=1$ | $y=y'=c_i$ | 58.6              | 34.4 | 6.3       | 50.6 | 39.7   | 53.9 |
> > |            | $y=c_i$    | 62.0              | 40.5 | 12.9      | 53.0 | 40.9   | 54.4 |
> > | $\delta=2$ | $y=y'=c_i$ | 44.7              | 20.0 | 2.5       | 26.7 | 34.6   | 44.7 |
> > |            | $y=c_i$    | 49.4              | 27.4 | 9.4       | 30.3 | 35.8   | 45.3 |
> > | $\delta=3$ | $y=y'=c_i$ | 31.6              | 14.4 | 1.3       | 13.0 | 28.2   | 31.7 |
> > |            | $y=c_i$    | 37.3              | 22.3 | 8.2       | 17.2 | 29.6   | 32.5 |
> >
> > However, we had not included these cases in the results to clearly distinguish from the bias due to counterfactual cases that are due to change in the decision.
> >
> > **Our results show that CPPBA consistently detects bias instances when RB (Nanda et al.) and CFA (Kusner et al.) do not.**
> >
> > As mentioned earlier, our experiments explicitly show settings where SP, EO, CFA, and RB all indicate very small disparities, yet CPPBA reveals large systematic confidence gaps between groups, even for fairness-aware models. These are cases where a model looks fair in terms of decisions and robustness but remains systematically more confident for some groups under small changes in non-sensitive features. Counterfactual fairness would call them fair because the label stays the same, and RB would view them as robust. CPPBA shows that there is still a structured formally-verifiable disparity.
> >
> > To illustrate this, we include several examples from our experiments. Considering Table 1 on the Adult dataset (gender, $\delta = 1$), CertiFair (100,50) shows almost no disparity in RB (95.6% vs. 96.5%) and only minor gaps in SP, EO, and CFA. Yet CPPBA uncovers a much larger confidence gap: 47.5% for Females versus 32.6\% for Males. Another example is in Table 2 on COMPAS, a fairness-agnostic model (500, 100) shows modest SP and EO disparity, while CPPBA reveals a wide 71.4% vs. 12.4% gap between Female and Male individuals.
> >
> > **Interpretation of the CPPBA objective**
> >
> > The restriction $\hat{y} = \hat{y}' = c_i$ is also what makes the CPPBA objective interpretable. Under that constraint, the expression in Equation (1) becomes exactly the log difference of the softmax margins between class $c_i$ and class $c_j$ for both inputs. Its sign therefore reflects how much additional margin the model assigns to the observed instance relative to its counterfactual across the entire perturbation region.  These cases matter in applications involving *ranking, triage, thresholding, or selective review*. A “positive but substantially less confident” output for one group can produce downstream disparities even though the predicted class is unchanged.

---

> > > ### Author Response · Authors · 2025-11-24
> > >
> > > >Equation 9 (which is unlabelled) is a logit difference interpreted as confidence. I am worried the resulting comparisons in the evaluation. If I recall correctly, CertiFair and Fair-N use additional constraints in their training objective. Thus, the logit magnitudes will change between comparable models, making comparison unfair. To further clarify, without temperature scaling or normalisation then I can't be sure the reported metrics are correct.
> > >
> > > >Equation (9) (the logit difference used as a “confidence” measure) appears to depend directly on raw logit magnitudes. Since models such as CertiFair and Fair-N are trained with additional constraints or penalties that alter the logit scale, how do you ensure that CPPBA scores remain comparable across models?
> > >
> > > Each CPPBA evaluation is performed on one model at a time for the original and counterfactual inputs (Figure 1). The CPPBA score for a subgroup is then computed as the proportion of inputs for which the verified objective is positive. This score depends only on the sign of the logit margin difference, not on the absolute scale of logits. Any uniform rescaling or shifting of logits within a model leaves the sign unchanged and therefore leaves the CPPBA score unchanged.
> > >
> > > In our experiments, for comparisons across different models, what we report are the CPPBA scores, which are the fraction of inputs in each subgroup that exhibit a confidence advantage or disadvantage relative to their counterfactuals. Since proportions reflect the distribution of verified positive versus negative outcomes, they remain comparable across architectures and training methods.

---

> > > > ### Comment · Reviewer_XuoK · 2025-11-25
> > > >
> > > > Thank you to the authors for taking time to discuss their paper. Please find below some furhter discussion.
> > > >
> > > > >The work is original. It is not a dual submission, and it does not reuse text or results from prior publications. Prior work on counterfactual fairness and robustness bias is fully cited, and we do not claim those ideas as new. We also note that the selections “no ethics review needed” and “research integrity issues” seem inconsistent, perhaps due to a clerical error. In case this is not an error, we expect a clear justification for such claims in the review, which we don’t seem to find for now.
> > > >
> > > > A: Apologies for this clerical issue. I have ammended my review, I did not mean to click "Research integrity" issues and apologise for this.
> > > >
> > > > > Nanda et al. check robustness only in a statistical sense, with no hard formal guarantees.
> > > >
> > > > A: In the submission itself (Sec. 3.4) the authors already characterise RB as using formal verification with provable guarantees, so the rebuttal’s claim that Nanda et al. offer only statistical robustness seems inaccurate.
> > > >
> > > > Furthermore, Nanda et al. explicitly use randomized smoothing to obtain per-point robustness certificates and describe these as provable defenses, providing robustness certificates for which inputs are provably robust within an $\ell_2$ ball. Thus it is not accurate to say that Nanda et al. only perform statistical robustness checks with no formal guarantees; they already rely on a certified robustness technique.
> > > >
> > > > While I agree your method allows assessment of entire regions, the claim above cannot be made.
> > > >
> > > > >The proposed method is highly derivative and therefore the novelty is low.
> > > >
> > > > A: I want to clarify my assessment of the methodological contribution. I agree that CP-PBA is not literally “CFA + RB stuck together”: the relational encoding that duplicates the network, couples $x$ and its counterfactual $x'$ , and enforces the same perturbation on both inputs is a small but genuine twist compared to running robustness analysis on a single trajectory. That is a reasonable design choice.
> > > >
> > > > However, this still looks like a fairly straightforward extension of existing verification machinery rather than any new work past small engineering additions. The fairness notion that ends up being certified is essentially a robust counterfactual margin difference. So while I acknowledge the relational formulation as an incremental refinement, I still view the overall methodological novelty as limited.
> > > >
> > > > > Our proposed framework can be readily adjusted to capture the cases where flipping the sensitive attribute changes the model’s decision.
> > > >
> > > > A: As written, CPPBDA requires $\hat{y} = \hat{y}' = c_i$. The paper’s current formulation, proofs, and discussion are anchored in the label-consistency setting.
> > > >
> > > > > We do not need to change our framework, only consider and feed the samples where flipping the sensitive attribute changes the model’s decision
> > > >
> > > > A: Simply feeding flip cases contradicts the paper’s constraint set and aggregation logic; you’d need a formal flip-aware spec and a revised aggregation rule before the results in Tables 1–3 remain interpretable, right?
> > > >
> > > > > CPPBA consistently detects bias instances when RB and CFA do not.
> > > >
> > > > A: CPPBA certifies a logit-margin property under shared perturbations and reports the share of positives. This demonstrates a different, confidence-based signal, not verified ground-truth “bias instances.” The larger gaps you highlight are also expected because CPPBA operates on pre-softmax margins, whose scale depends on training and calibration; without normalization, cross-model differences can reflect logit scale rather than fairness. A more accurate wording is that CPPBA is complementary to SP/EO/CFA/RB and reveals additional confidence disparities.

---

> > > > > ### Comment · Reviewer_XuoK · 2025-11-25
> > > > >
> > > > > > Interpretation of the CPPBA objective
> > > > >
> > > > > A: Your claim that the $\hat{y} = \hat{y}' = c_i$ restriction “makes the objective interpretable” is only partially persuasive in it's current form. It yields a logit/softmax margin-difference conditional on label agreement, but the resulting number is scale- and calibration-dependent, varies with the choice of comparison class $c_j$ in the multi-class case, and is not invariant across models with different training penalties. Without temperature scaling or probability-space calibration and a within-model analysis, the larger gaps you report can conflate logit scale with fairness.
> > > > >
> > > > > > This score depends only on the sign of the logit margin difference, not on the absolute scale of logits.
> > > > >
> > > > > A: Your “sign-only” defense doesn’t establish cross-model comparability: changes in training typically alter margins non-uniformly across inputs/classes, so the sign of the logit-margin difference can flip for calibration reasons unrelated to fairness; moreover, your paper notes that verification can be inconclusive and coverage worsens with larger $\delta$ (Section 3.5 and 3.6.1), so the reported proportions are conditional on the verified subset, which can differ by subgroup/model. Finally, as you state before Table 1, different networks are used for different attributes, and relaxation tightness varies with architecture, each of these factors can shift certified signs independently of any substantive bias.
> > > > >
> > > > > I appreciate the authors discussing their paper and my review in more detail. In light, I have updated my review, and score, and would be more than happy to continue discussion.

---

> ### Author Response · Authors · 2025-11-27
>
> We thank the reviewer for their response and clarification/amending their review concerning the research integrity flag.
>
> >Nanda et al. check robustness only in a statistical sense, with no **hard** formal guarantees.
>
> We would like to emphasize the wording “hard” (hard as opposed to soft, e.g., in soft margins). As we understand, Nanda et al. do not provide hard formal guarantees as they also mention in their paper in the section they use randomized smoothing: “Randomized smoothing transforms the base classifier $f$ to a new smooth classifier $g$ by averaging the output of $f$ over noisy versions of $x$. This new classifier $g$ is more robust to perturbations while also having accuracy on par to the original classifier. It is also possible to calculate the radius $\delta_x$ (in the $\ell_2$ distance) in which, with high probability, a given input's prediction remains the same for the smoothed classifier (i.e.\ $d_{\theta}(x) \ge \delta_x$). A given input $x$ is then said to be provably robust, **with high probability**, for a $\delta_x$ $\ell_{2}$-perturbation where $\delta_x$ is the robustness certificate of $x$.” Therefore, our understanding is that Nanda et al. can only provide soft/probabilistic guarantees.
>
> As we write in our paper, however, to make a fair comparison between RB and our approach in the experiments, “Unlike attack-dependent empirical adversarial accuracy, we quantify RB using formal verification methods that provide provable robustness guarantees, enabling reliable group-wise assessment where differences in robustness indicate potential unfairness in the model's decision boundary.”
>
> > The proposed method is highly derivative and therefore the novelty is low.
>
> > However, this still looks like a fairly straightforward extension of existing verification machinery rather than any new work past small engineering additions. The fairness notion that ends up being certified is essentially a robust counterfactual margin difference. So while I acknowledge the relational formulation as an incremental refinement, I still view the overall methodological novelty as limited.
>
> We would like to summarize our contribution:
>
> 1) From a conceptual point of view, the coupling between the input and its counterfactual with the same perturbation across an entire region brings about a new fairness concept. In essence, we consider the exact same perturbation (e.g., the vector that captures 2 years older and 10% higher salary) for a male and to its female counterfactual, to investigate how the same change affects the decision outcome for a male compared to a female. And, we do this for all points in the entire neighborhood of the input and its counterfactual.
>
> 2) From a technical point of view, as discussed above, we provide a *sound* formulation with *hard* formal guarantees over *an entire region*, as opposed to other techniques. Formulating the problem in a sound fashion in the presence of non-linear softmax functions, without introducing additional over-approximation is also one of our contributions.
>
> 3) From an experimental point of view, we would like to reiterate that our approach identifies instances of bias that the other approaches do not, as Tables 1, 2, 3 show.  Considering Table 1 on the Adult dataset (gender, $\delta = 1$), CertiFair (100,50) shows almost no disparity in RB (95.6% vs. 96.5%) and only minor gaps in SP, EO, and CFA. Yet CPPBA uncovers a much larger confidence gap: 47.5% for Females versus 32.6% for Males. Another example is in Table 2 on COMPAS, a fairness-agnostic model (500, 100) shows modest SP and EO disparity, while CPPBA reveals a wide 71.4% vs. 12.4% gap between Female and Male individuals.

---

> ### Author Response · Authors · 2025-11-27
>
> >A: As written, CPPBDA requires $\hat{y} = \hat{y}' = c_i$. The paper’s current formulation, proofs, and discussion are anchored in the label-consistency setting.
>
> >A: Simply feeding flip cases contradicts the paper’s constraint set and aggregation logic; you’d need a formal flip-aware spec and a revised aggregation rule before the results in Tables 1–3 remain interpretable, right?
>
> We thank the reviewer for raising this point and we will clarify and address this unclarity in the next revised version. We would like to clarify that the condition purpose is mainly for clarification (there are no variables in that condition). It was introduced to highlight that we consider only the ones that the input and counterfactual are both correctly classified. We could simply consider the ones that the input is correctly classified (counterfactual can have correct or incorrect decisions) by feeding those samples to the optimization problem. Relaxing it does not affect the results, proofs, and our contribution.
>
> *In essence, we calculate the minimum value of $(x_i - x_j) - (x'_i - x'_j)$ in the Optimization Problem Objective Function (1). A positive value for the objective function means that the input is more biased towards the correct class than its counterfactual, in the entire region. In other words, a positive objective function means $(x_i - x_j) - (x'_i - x'_j)>0$, that is $(x_i - x_j) >(x'_i - x'_j)$. This means that $x_i$ relative to $x_j$ is larger than $x'_i$ relative to $x'_j$.*
>
> >A: CPPBA certifies a logit-margin property under shared perturbations and reports the share of positives. This demonstrates a different, confidence-based signal, not verified ground-truth “bias instances.” The larger gaps you highlight are also expected because CPPBA operates on pre-softmax margins, whose scale depends on training and calibration; without normalization, cross-model differences can reflect logit scale rather than fairness. A more accurate wording is that CPPBA is complementary to SP/EO/CFA/RB and reveals additional confidence disparities.
>
> We agree with the wording of the reviewer that CPPBA is complementary to SP/EO/CFA/RB.
>
> At the same time, we also believe that CPPBA shows verified ground-truth “bias instances”. *If our optimization problem has a positive solution (a solution that leads to positive objective function value), it essentially means that the input is more biased towards the correct class than its counterfactual, in the entire region. And, this is guaranteed to hold, due to the formal nature of our approach and its soundness property.* In our understanding, this is clearly an instance of persistent bias. Formally, as mentioned above, a positive objective function means $(x_i - x_j) - (x'_i - x'_j)>0$, which is equivalent to $(x_i - x_j) >(x'_i - x'_j)$. This means that $x_i$ relative to $x_j$ is larger than $x'_i$ relative to $x'_j$, which highlights the bias of the original input towards the correct class, compared to its counterfactual.
>
> Finally, we would also like to highlight that CPPBA actually works on post-softmax margins, which are normalized values. *The softmax non-linearities are exactly (without introducing any additional over-approximation) peeled thanks to our formulation that considers the relative margins. Please see the Equation in Lines 134-137. This formulation allows the objective to be linear, enabling the adoption of efficient linear programming solvers.*
>
> >A: Your claim that the $\hat{y} = \hat{y}' = c_i$ restriction “makes the objective interpretable” is only partially persuasive in its current form. It yields a logit/softmax margin-difference conditional on label agreement, but the resulting number is scale- and calibration-dependent, varies with the choice of $c_j$ comparison class in the multi-class case, and is not invariant across models with different training penalties. Without temperature scaling or probability-space calibration and a within-model analysis, the larger gaps you report can conflate logit scale with fairness.
>
> While we’re not quite sure what the main source of the question is, we’d be happy to provide more detail. As we mentioned in our earlier response, we assume both models in Figure 1 are the same, therefore, we mainly work in a within-model regime. At the same time, our formulation is post-softmax and hence works on normalized values. Finally, the equation in Lines 134-137 is invariant to temperature scaling, due to its relational nature.
>
> When reporting the evaluation results for different models (in Tables, not having two different models in Figure 1), the interpretation is not that “model A has larger margins than model B,” but that “model A yields a higher fraction of inputs from a given subgroup (for example male) that show a locally-persistent bias [toward the correct class] over their counterfactuals (female)”.

---

> ### Author Response · Authors · 2025-11-27
>
> >A: Your “sign-only” defense doesn’t establish cross-model comparability: changes in training typically alter margins non-uniformly across inputs/classes, so the sign of the logit-margin difference can flip for calibration reasons unrelated to fairness; moreover, your paper notes that verification can be inconclusive and coverage worsens with larger $\delta$ (Section 3.5 and 3.6.1), so the reported proportions are conditional on the verified subset, which can differ by subgroup/model. Finally, as you state before Table 1, different networks are used for different attributes, and relaxation tightness varies with architecture, each of these factors can shift certified signs independently of any substantive bias.
>
> We would like to emphasize that we do not claim that margins themselves are comparable across models; CPPBA reports the fraction of inputs whose input-vs-counterfactual margin-difference is provably positive over the whole perturbation region. This proportion does not depend on the absolute scale of logits. It captures how often each model yields a persistent bias [toward the correct class] for one subgroup (e.g., male) over its counterfactual (e.g., female) inside a shared local neighborhood.
>
> We acknowledge, however, that the verified cases might be affected for different models, because of the verification techniques. However, to address this, our experiments evaluate two complementary scenarios: an observed scenario and a flipped scenario. In the observed scenario, we take inputs from a subgroup (e.g., females) and their counterfactuals, and we check whether CPPBA can be verified for the actual–counterfactual comparison. In the flipped scenario, we evaluate CPPBA again, but now using the counterfactuals (e.g., males obtained by flipping females) as the reference and comparing them against the original inputs (counterfactual–actual). In this setup, the sum of “female observed” and “male flipped” CPPBA values should be 100% if and only if 1) no cases are lost due to verifier over-approximation, and 2) for every input in the dataset, the property holds across the entire perturbation region in one of the two directions (male-flipped or female-observed). As our models are relatively small, it is more likely that deviations from 100% arise from the second reason. Here is an example from the paper:
>
> In the COMPAS dataset with the (200, 100) model, for $\delta=1$, CertiFair yields CPPBA values of 6.3% for females and 50.6% for males in the observed setting. In the flipped setting, the corresponding CPPBA values are 14.2% and 55.7% for females and males, respectively. Summing the complementary cases gives 6.3+55.7=62.0 and 50.6+14.2=64.8, which are both well below 100. This could be due to natural inconsistency within the perturbation region being explored or limitations of the verification tool. However, this issue does not appear for Fair-N. Under the same model and perturbation, Fair-N attains CPPBA values of 39.7% and 53.9% for females and males in the observed setting, and 38.9% and 55.8% in the flipped setting (39.7+55.8=95.5 and 53.9+38.9=92.8).
>
> In summary, we compare subgroups within the same trained model and interpret cross-model differences. When comparing different training methods, the point is to evaluate whether a given training scheme reduces the frequency with which it produces locally persistent disparities, as measured by a sound verifier.
>
> We thank the reviewer for their time and effort and for participating in the discussion. We'd appreciate knowing if our responses were satisfactory and if our clarified perspective might improve your evaluation.

---

### Official Review · Reviewer_GqzJ · 2025-10-18

**Soundness:** 3
**Presentation:** 3
**Contribution:** 2
**Rating:** 4
**Confidence:** 4

**Summary:**

The paper introduces a new fairness metric called Ceteris Paribus Persistent-Bias-Aware (PBA) Fairness. The core idea is to detect subtle biases in a model's confidence, even when its final predictions for different groups seem fair. The authors test PBA on standard fairness datasets (Adult, COMPAS, German Credit) and show that it successfully identifies these confidence-based disparities, which are often overlooked by metrics like Statistical Parity and Equalized Odds.

**Strengths:**

1. Individual fairness is an important topic in fairness studies.

2.  The overall presentation is smooth and easy to follow.

3. Extensive experiments are presented.

**Weaknesses:**

1. While the paper presents an interesting approach, the novelty of the contribution appears incremental. The proposed PBA fairness metric can be viewed as a synthesis of existing concepts, primarily extending Counterfactual Fairness (CFA) by incorporating a confidence-based analysis within a perturbed neighborhood. As the paper's own related work section suggests, using confidence scores to uncover subtle biases is not a new idea, which makes the overall contribution feel more like a refinement than a foundational shift.

2. The authors rightly motivate their work by highlighting the scalability issues of prior methods in high-dimensional settings. However, the evaluation is confined to small, fully-connected networks on low-dimensional tabular data. The paper does not provide evidence that the proposed method, which relies on linear relaxation, would itself scale to the complex architectures (e.g., CNNs) and high-dimensional inputs (e.g., images).

3. The practical application of the method is limited by the lack of clear guidance on selecting the perturbation bound, $\delta$. This hyperparameter is critical as it defines the "local neighborhood" for verification. While the paper uses fixed values, it does not offer a principled process for how a practitioner should select an appropriate $\delta$ for a new dataset or model, which could impact the reliability and comparability of the results.

4. A significant methodological concern is the explicit exclusion of counterfactuals that result in a class change ($\hat{y} \ne \hat{y}'$). By design, the metric overlooks a critical form of unfairness related to model robustness. For instance, a prediction for one demographic group might be fragile and easily flipped by minor perturbations, while remaining stable for another group. This disparity in robustness is a crucial fairness issue that the current formulation cannot capture.

**Questions:**

1. The final metric is reported as the proportion of inputs exhibiting bias, which treats all instances of unfairness equally, regardless of magnitude. This raises the question: Is a model with a widespread, low-magnitude bias (e.g., 90% of inputs show a tiny confidence gap) less fair than a model with a concentrated, high-magnitude bias (e.g., 10% of inputs show a massive gap)?

---

> ### Author Response · Authors · 2025-11-24
>
> We thank the reviewer for the thoughtful comments and for highlighting the strengths of the paper. Below we address the concerns raised.
>
> > … the novelty of the contribution appears incremental…
>
> We appreciate this perspective and acknowledge that CPPBA builds on prior concepts such as counterfactual reasoning and local robustness. Its contribution is not to introduce a new primitive, but to formalize and verify a new relational fairness property. CPPBA couples an input and its counterfactual under identical perturbations, which makes it possible to compare their margins within one shared region. The objective represents a locally-persistent confidence gap since the model consistently favors one version of the individual across all perturbations in that region. Prior confidence-based fairness work typically evaluates confidence only pointwise or does not impose a shared perturbed region for both inputs. The experiments show that this relational property exposes disparities that are not captured by SP, EO, CFA, or robustness-based metrics. To illustrate this, we include several examples from our experiments. Considering Table 1 on the Adult dataset (gender, $\delta = 1$), CertiFair (100, 50) shows almost no disparity in RB (95.6% vs. 96.5%) and only minor gaps in SP, EO, and CFA. Yet CPPBA uncovers a much larger confidence gap: 47.5% for Females versus 32.6% for Males. Another example is in Table 2 on COMPAS, a fairness-agnostic model (500, 100) shows modest SP and EO disparity, while CPPBA reveals a wide 71.4% vs. 12.4% gap between Female and Male individuals.
>
> > … would itself scale to the complex architectures (e.g., CNNs) and high-dimensional inputs (e.g., images)
>
> In the current paper, we intentionally adopted the same model families used in prior fairness-verification studies (e.g., Fair-N published in AAAI/ACM AIES 2021 and CertiFair published in AAAI 2023), all of which rely on small fully connected networks for tabular benchmarks. This allows for a direct comparison to established baselines and ensures that the verification component remains tractable.
>
> Conceptually, CPPBA is model-agnostic because it only requires a mechanism to determine the sign of the relational confidence margin between an input and its counterfactual. Any verification backend that can verify whether the margin is positive can be used to evaluate CPPBA. For CNNs or image inputs, the same principle applies where one specifies a perturbation region over pixels and uses the chosen verification method to evaluate the CPPBA relational objective. The extension is therefore straightforward in principle.
>
> > lack of clear guidance on selecting the perturbation bound
>
> The perturbation radius determines the size of the local region explored around each input and its counterfactual. In practice, $\epsilon$ is chosen to reflect meaningful variation in the non-sensitive features. In our experiments, we selected $\epsilon$ by examining the range of the numeric attributes in each dataset, choosing values that represent small but meaningful changes relative to those ranges. Using this feature-aware selection ensured that the CPPBA and RB neighborhoods were comparable in spirit, so any differences between the two metrics reflect their underlying fairness notions rather than mismatched perturbations.
>
> In the German Credit dataset, the numerical attributes subject to perturbation are *duration* and *amount*. Duration spans 4-72 months (range 68), and amount ranges from 250 to 18,424 (range 18,174). A unit perturbation corresponds to $3/68$ for duration and $1000/18,174$ for amount. These perturbations are applied in normalized space and clipped to the valid $[0,1]$ interval, ensuring that values never fall below the minimum or exceed the maximum. This feature-aware scaling keeps all perturbed points within realistic regions of the input space and ensures that CPPBA explores semantically coherent local neighborhoods around each sample.

---

> > ### Author Response · Authors · 2025-11-24
> >
> > > the metric overlooks a critical form of unfairness related to model robustness
> >
> > Class-flip counterfactuals are already handled by CFA and robustness-based fairness metrics. CPPBA is designed to study a different phenomenon by isolating confidence asymmetry in cases where the predicted class remains the same. These two perspectives are complementary where CFA identifies decision-level vulnerabilities, while CPPBA reveals systematic confidence gaps that persist even when decisions are stable. Our goal is not to replace robustness-based approaches, but to highlight a form of disparity they cannot detect. Such cases matter in applications involving *ranking, triage, thresholding, or selective review*. A “positive but substantially less confident” output for one group can produce downstream disparities even though the predicted class is unchanged.
> >
> > That said, we *can* consider class-flip counterfactuals by considering and feeding the samples where flipping the sensitive attribute changes the model’s decision. As an example, we examined the results for the COMPAS dataset using the (200, 100) model.
> >
> > |            |            | Fairness-agnostic |      | CertiFair |      | FairN  |      |
> > |------------|------------|-------------------|------|-----------|------|--------|------|
> > |            |            | Female            | Male | Female    | Male | Female | Male |
> > | $\delta=1$ | $y=y'=c_i$ | 58.6              | 34.4 | 6.3       | 50.6 | 39.7   | 53.9 |
> > |            | $y=c_i$    | 62.0              | 40.5 | 12.9      | 53.0 | 40.9   | 54.4 |
> > | $\delta=2$ | $y=y'=c_i$ | 44.7              | 20.0 | 2.5       | 26.7 | 34.6   | 44.7 |
> > |            | $y=c_i$    | 49.4              | 27.4 | 9.4       | 30.3 | 35.8   | 45.3 |
> > | $\delta=3$ | $y=y'=c_i$ | 31.6              | 14.4 | 1.3       | 13.0 | 28.2   | 31.7 |
> > |            | $y=c_i$    | 37.3              | 22.3 | 8.2       | 17.2 | 29.6   | 32.5 |
> >
> > However, we had not included these cases in the results to clearly distinguish from the bias due to counterfactual cases that are due to change in the decision.
> >
> > > Is a model with a widespread, low-magnitude bias (e.g., 90% of inputs show a tiny confidence gap) less fair than a model with a concentrated, high-magnitude bias (e.g., 10% of inputs show a massive gap)?
> >
> > CPPBA reports the proportion of inputs with a positive relational margin because the sign of this margin is the reliable part of the verification result. A positive sign certifies that the model consistently favors the original input over its counterfactual across the entire perturbation region. In contrast, the magnitude of the verified margin is less meaningful as it can change with logit scaling, even when the underlying model behaviour is the same. For this reason, CPPBA is designed to measure whether persistent bias occurs, not how large the relaxed numerical margin happens to be. While it would be possible to report margin magnitudes, doing so would be less informative and potentially misleading compared with focusing on the sign.

---

> ### Comment · Reviewer_GqzJ · 2025-11-25
>
> Thanks for the reply. However, the proposed concerns have not been thoroughly resolved. Moreover, as pointed out by other reviewers/during discussions, some issues still need to be further clarified. Therefore, I would like to maintain my original score.

---

> ### Author Response · Authors · 2025-11-27
>
> We would appreciate knowing which of our responses have not been satisfactory and what is the underlying reasons, e.g., why our clarification is not sufficient to address the reviewer's concern. Perhaps we can substantiate our response or further clarify our perspective to address reviewer's concerns, which is the idea with the discussion phase. This will also help us revise and improve our work in the future. Thank you in advance.

---

### Official Review · Reviewer_fiiY · 2025-10-30

**Soundness:** 3
**Presentation:** 3
**Contribution:** 3
**Rating:** 4
**Confidence:** 4

**Summary:**

The paper proposes PBA fairness, a metric that detects confidence-based bias: when a model gives the same prediction to an input and its counterfactual (flipped sensitive attribute), but with different confidence. It computes the worst-case confidence gap in a locally perturbed neighborhood, isolating the effect of the sensitive attribute.

**Strengths:**

The paper presents an original fairness perspective by focusing on confidence disparities rather than prediction labels as traditionally done. The method is formally defined and clearly explained, and the comparison with existing fairness metrics across multiple datasets is well-executed.

**Weaknesses:**

- The method flips the sensitive attribute s while keeping the other features z fixed and applying the same local perturbations to the numerical parts of z, which isolates the marginal effect of s. However, this can produce causally implausible counterfactuals when s influences aspects of z (e.g., age affecting credit-history length). As a result, the measured confidence gaps may not reflect meaningful unfairness, but rather effects arising from counterfactuals that do not correspond to any plausible individual. Without modeling these causal links, it is unclear whether high PBA scores indicate unfair outcomes or simply unrealistic interventions.
- PBA is defined only on non-flip cases, so it evaluates a restricted subset of the data and excludes instances where flipping the sensitive attribute changes the decision. When that subset is small, a high PBA mainly reflects behavior on a narrow slice, making standalone interpretation uncertain.
- The paper does not fully justify why confidence bias matters, especially if the final decisions are equal. It would benefit from concrete examples or applications where differences in model confidence lead to tangible downstream impacts.
- The paper only detects confidence bias, but doesn’t propose a way to reduce it.

**Questions:**

1) How do you ensure that flipped counterfactuals correspond to plausible individuals, rather than off-manifold inputs that could inflate the confidence gaps?
2) How do you account for the metric’s coverage (since PBA is computed only on non-flip cases)? What proportion of the dataset is excluded due to decision flips, and how should PBA scores be interpreted when coverage is low?

---

> ### Author Response · Authors · 2025-11-27
>
> We thank the reviewer for the thoughtful and constructive feedback. Below we address the concerns in detail.
>
> > … Without modeling these causal links, it is unclear whether high PBA scores indicate unfair outcomes or simply unrealistic interventions.
>
> > How do you ensure that flipped counterfactuals correspond to plausible individuals
>
> We appreciate this concern and agree that causal validity is an important consideration. CPPBA follows the widely used attribute-swap counterfactual assumption from counterfactual fairness work (Kusner et. al. NeurIPS 2017), where only the sensitive attribute is intervened on. The goal is to examine the model’s local behaviour under a minimal and controlled change rather than to reconstruct a fully causally accurate profile. This intervention style is standard in fairness methods, including CertiFair (AAAI 2023) and subsequent work on relational verification such as the confidence-based two-safety checks of Athavale et al. (CAV 2024).
>
> For example, in the Adult and COMPAS datasets, flipping the sensitive attribute (gender or race) does not violate any structural constraints, and the remaining attributes encode socioeconomic or behavioural information that can co-occur with any value of the sensitive attribute. These datasets do not include hard logical or causal dependencies that would render a swapped counterfactual infeasible.
>
> CPPBA also reduces off-manifold risk by keeping all perturbations small and by applying them symmetrically to both $x$ and $x'$. This ensures that both inputs remain within a tightly bounded neighbourhood around the original sample and avoids drifting into unrealistic regions of the input space. The method therefore compares how the model treats two nearly identical trajectories instead of relying on large or implausible counterfactual jumps.
>
> > PBA is defined only on non-flip cases…
>
> It is correct that CPPBA is defined only on non-flip instances. This is intentional because CPPBA is designed to measure a different phenomenon from decision-flip unfairness. Decision flips are already captured by counterfactual fairness metrics and by robustness-based approaches. CPPBA instead focuses on confidence asymmetry in cases where the predicted class is the same. A CPPBA score should therefore be interpreted as the proportion of inputs that keep the same label after the sensitive-attribute flip and that still exhibit a persistent confidence difference across the local region. This separates confidence-based effects from decision-boundary sensitivity. When the number of non-flip cases is small, CPPBA represents behaviour on a specific subset of inputs where confidence differences can still produce downstream impacts even though the top prediction does not change.
>
> That said, we *can* consider class-flip counterfactuals by considering and feeding the samples where flipping the sensitive attribute changes the model’s decision. As an example, we examined the results for the COMPAS dataset using the (200, 100) model.
>
> |            |            | Fairness-agnostic |      | CertiFair |      | FairN  |      |
> |------------|------------|-------------------|------|-----------|------|--------|------|
> |            |            | Female            | Male | Female    | Male | Female | Male |
> | $\delta=1$ | $y=y'=c_i$ | 58.6              | 34.4 | 6.3       | 50.6 | 39.7   | 53.9 |
> |            | $y=c_i$    | 62.0              | 40.5 | 12.9      | 53.0 | 40.9   | 54.4 |
> | $\delta=2$ | $y=y'=c_i$ | 44.7              | 20.0 | 2.5       | 26.7 | 34.6   | 44.7 |
> |            | $y=c_i$    | 49.4              | 27.4 | 9.4       | 30.3 | 35.8   | 45.3 |
> | $\delta=3$ | $y=y'=c_i$ | 31.6              | 14.4 | 1.3       | 13.0 | 28.2   | 31.7 |
> |            | $y=c_i$    | 37.3              | 22.3 | 8.2       | 17.2 | 29.6   | 32.5 |
>
> However, we had not included these cases in the results to clearly distinguish from the bias due to counterfactual cases that are due to change in the decision.

---

> > ### Author Response · Authors · 2025-11-27
> >
> > > How do you account for the metric’s coverage (since PBA is computed only on non-flip cases)? What proportion of the dataset is excluded due to decision flips, and how should PBA scores be interpreted when coverage is low?
> >
> > To address the question about coverage, we report the proportion of non-flip cases for the COMPAS experiments where gender is the sensitive attribute and the (200, 100) model architecture. For the FairN model, 785 of 793 correctly classified samples remain in the same correctly predicted class, which corresponds to 99.0%. For the CertiFair model, 787 of 827 correctly classified samples remain in the same class, corresponding to 95.2%. For the fairness-agnostic model, 757 of 833 correctly classified samples remain in the same class, corresponding to 90.9%. These results show that CPPBA evaluates a substantial portion of the dataset across all three models. This is especially the case because we restrict attention to *correctly classified samples* and, *within that set, only to those that keep the same correct class after the sensitive-attribute flip*. However, a CPPBA score should be interpreted as the prevalence of persistent confidence asymmetry within this subset, that is, among inputs whose predicted class remains the same after the sensitive-attribute flip. When coverage is lower, the metric characterises behaviour on this specific group of cases where confidence differences can influence downstream processing even when the final prediction agrees.
> >
> > > … It would benefit from concrete examples or applications where differences in model confidence lead to tangible downstream impacts.
> >
> > This is an important point, and we are happy to clarify. Confidence differences affect systems in several realistic settings:
> >
> > * Ranking or prioritization (e.g., loan applicants sorted by confidence rather than class).
> > * Selective prediction and uncertainty-based abstention, where low confidence triggers human review.
> > * Threshold-based decision policies, where the final decision depends not only on the top class but on margin relative to a calibrated threshold.
> > * Audit settings, where confidence disparities can indicate latent structural bias even when labels match.
> >
> > In all of these cases, two individuals with the same predicted class but systematically different confidence can be treated differently downstream. CPPBA aims to uncover precisely this hidden persistent asymmetry.
> >
> > > The paper only detects confidence bias, but doesn’t propose a way to reduce it.
> >
> > We agree that mitigation is an important next step. Our contribution in this paper has been to formulate CPPBA as a sound problem in formal methods. CPPBA is designed as a diagnostic and auditing tool, similar to CFA and robustness-bias metrics in Nanda et al. Extending CPPBA into a training objective (e.g., by regularizing the relational margin) is feasible and part of our ongoing work.
> >
> > We thank the reviewer for their time and effort. We'd appreciate knowing if our responses were satisfactory and if our clarified perspective might improve your evaluation.

---

### Official Review · Reviewer_oGie · 2025-11-01

**Soundness:** 2
**Presentation:** 3
**Contribution:** 2
**Rating:** 4
**Confidence:** 3

**Summary:**

This paper introduces Ceteris Paribus Persistent-Bias-Aware (PBA) fairness, a new formalism for measuring fairness in neural networks. Unlike existing group-level metrics or individual fairness metrics that rely on predefined similarity measures, PBA fairness quantifies local, confidence-based bias by comparing a model’s confidence between an input and its counterfactual, while jointly perturbing all numerical features within a bounded neighborhood. The authors show that this method identifies persistent confidence disparities overlooked by classical fairness metrics, using formal verification and linear relaxations for ReLU networks. Experiments on Adult, COMPAS, and German Credit datasets demonstrate that PBA fairness reveals hidden local biases, even in fairness-aware models.

**Strengths:**

1. PBA fairness identifies confidence-level disparities that standard metrics miss. The introduction of locally persistent bias offers an interpretable lens to analyze fairness beyond label-level metrics, contributing to nuanced fairness assessment.
2. The study includes three major datasets and compares fairness-aware vs. agnostic training under multiple metrics.
3. The notations and the mathematical formulation are clear and consistent.
4. The writing of the paper is easy to follow. The presentation/visualization of the introduced metric and the experimental results is very clear.

**Weaknesses:**

1. My biggest concern lies in the experimental design. The current experiments are confined to small feed-forward networks. There’s no indication of feasibility for modern deep architectures or larger inputs.
2. The current experiments are limited to small tabular datasets. Larger datasets and data with other modalities should be tested. In addition, the used Adult dataset is outdated [1].
3. The compared fairness-aware baselines are limited. There has been a surge of fair ML research recently. Is there any reason to select CertiFair and Fair-N, and only the two?
4. In the presentation of the results, an additional column showing the difference between the two groups would make it easier for readers to compare. Also, the results would be more convincing with the standard deviation.
5. There is limited discussion on the real-world impact or interpretive significance of the confidence disparities uncovered.


[1] Ding, Frances, et al. "Retiring adult: New datasets for fair machine learning." Advances in neural information processing systems 34 (2021): 6478-6490.

**Questions:**

1. See my questions in Weaknesses.
2. How sensitive is PBA fairness to the choice of perturbation bounds? Is there any suggestions on how to select proper perturbation bounds in your method?
3. How do PBA fairness disparities correlate with real-world outcomes? Is there any example where high-confidence unfair regions are also where models make most errors?

---

> ### Author Response · Authors · 2025-11-27
>
> We thank the reviewer for the constructive feedback and the positive comments. We address the concerns below.
>
> > There’s no indication of feasibility for modern deep architectures or larger inputs.
>
> > The current experiments are limited to small tabular datasets.
>
> CPPBA itself does not rely on any specific architecture or data modality. The relational objective only requires upper and lower bounds on neuron activations under perturbations, which can in principle be propagated through convolutional layers or deeper networks. However, the verification back end that certifies these bounds currently scales reliably only to low and medium dimensional tabular models. This constraint is shared with existing fairness-verification methods such as Fair-N (AAAI/ACM AIES 2021) and CertiFair (AAAI 2023).
>
> For this reason our experiments focus on tabular datasets and small feed-forward networks, which allow complete and certified evaluation of CPPBA as well as SP, EO, CFA, and robustness-based metrics. Datasets such as Adult, COMPAS, and German Credit remain standard benchmarks in the fairness verification literature for exactly this reason. As verification tools evolve, applying CPPBA to larger models and datasets is a natural extension.
>
> > The compared fairness-aware baselines are limited.
>
> We used CertiFair(AAAI 2023) and Fair-N (AAAI/ACM AIES 2021) because they are fairness-aware training methods designed for tabular settings and are implemented on the same class of models used in our experiments. This allows all fairness metrics, including CPPBA, CFA, and robustness-based fairness, to be evaluated under identical modeling and training conditions. Using these baselines ensures that differences come from the fairness notions themselves rather than from mismatched architectures or training procedures.
>
> > In the presentation of the results, an additional column showing the difference between the two groups would make it easier for readers to compare. Also, the results would be more convincing with the standard deviation.
>
> We appreciate the suggestion to include explicit group difference columns and standard deviations. In the current version we already highlight the group disparities using gray shading in the table. The shading reflects the magnitude of the subgroup difference with lighter tones indicating smaller gaps and darker tones marking larger disparities. This visual choice keeps the table readable without adding extra columns that would make the layout more difficult to follow. Regarding standard deviations, the reported values are exact proportions. For each metric we count how many inputs satisfy the certified property and divide by the size of the relevant subset. This process is entirely deterministic and does not involve sampling or repeated stochastic runs. As a result, there is no variability across trials and therefore no meaningful standard deviation to report.
>
>  > There is limited discussion on the real-world impact or interpretive significance of the confidence disparities uncovered.
>
> Confidence differences matter in practical settings where decisions depend not only on the final predicted class but on the model’s certainty. This includes ranking and prioritization systems, selective prediction and abstention, dynamic thresholding policies, and human-in-the-loop pipelines where low confidence triggers manual review. CPPBA examines cases where two individuals have the same predicted label but systematically different confidence across the local neighbourhood. These differences can produce meaningful downstream effects even when predictions match.  However, CFA cannot capture the same disparities. For instance, in Table 1 on the Adult dataset (gender, $\delta = 1$), CertiFair (100, 50) reports only minor gaps under CFA, yet CPPBA reveals a substantially larger confidence difference which is 47.5% for females compared to 32.6% for males.

---

> > ### Author Response · Authors · 2025-11-27
> >
> > > How sensitive is PBA fairness to the choice of perturbation bounds? Is there any suggestions on how to select proper perturbation bounds in your method?
> >
> > The perturbation bound defines the size of the local region in which CPPBA evaluates persistent bias. In practice, it can be chosen using the range of numerical features. In our experiments we followed standard practice from robustness verification and selected small values that represent plausible variation in the tabular attributes.
> >
> > In the German Credit dataset, the numerical attributes subject to perturbation are *duration* and *amount*. Duration spans 4-72 months (range 68), and amount ranges from 250 to 18,424 (range 18,174). A unit perturbation corresponds to $3/68$ for duration and $1000/18,174$ for amount. These perturbations are applied in normalized space and clipped to the valid $[0,1]$ interval, ensuring that values never fall below the minimum or exceed the maximum. This feature-aware scaling keeps all perturbed points within realistic regions of the input space and ensures that CPPBA explores semantically coherent local neighborhoods around each sample.
> >
> > > How do PBA fairness disparities correlate with real-world outcomes? Is there any example where high-confidence unfair regions are also where models make most errors?
> >
> > In the current work CPPBA is intentionally applied only to correctly classified inputs. This isolates confidence asymmetry from decision-level errors and allows us to study a phenomenon that is not captured by accuracy, false negatives, or other error-based metrics. For this reason the PBA score is not designed to track misclassification behaviour directly.
> >
> > That said, we *can* consider class-flip counterfactuals by considering and feeding the samples where flipping the sensitive attribute changes the model’s decision. As an example, we examined the results for the COMPAS dataset using the (200, 100) model.
> >
> > |            |            | Fairness-agnostic |      | CertiFair |      | FairN  |      |
> > |------------|------------|-------------------|------|-----------|------|--------|------|
> > |            |            | Female            | Male | Female    | Male | Female | Male |
> > | $\delta=1$ | $y=y'=c_i$ | 58.6              | 34.4 | 6.3       | 50.6 | 39.7   | 53.9 |
> > |            | $y=c_i$    | 62.0              | 40.5 | 12.9      | 53.0 | 40.9   | 54.4 |
> > | $\delta=2$ | $y=y'=c_i$ | 44.7              | 20.0 | 2.5       | 26.7 | 34.6   | 44.7 |
> > |            | $y=c_i$    | 49.4              | 27.4 | 9.4       | 30.3 | 35.8   | 45.3 |
> > | $\delta=3$ | $y=y'=c_i$ | 31.6              | 14.4 | 1.3       | 13.0 | 28.2   | 31.7 |
> > |            | $y=c_i$    | 37.3              | 22.3 | 8.2       | 17.2 | 29.6   | 32.5 |
> >
> > However, we had not included these cases in the results to clearly distinguish from the bias due to counterfactual cases that are due to change in the decision.
> >
> > We thank the reviewer for their time and effort. We'd appreciate knowing if our responses were satisfactory and if our clarified perspective might improve your evaluation.

---

### Author Response · Authors · 2025-12-03

Dear Area Chair,

Thank you for taking the time to review this message and for your effort toward a fair evaluation process.

We have received an integrity flag by mistake from Reviewer XuoK for our submission 18726: *Flag For Ethics Review: No ethics review needed., Yes, Research integrity issues (e.g., plagiarism, dual submission)*

The reviewer has indicated that this has been a mistake and removed the flag:
*Apologies for this clerical issue. I have amended my review, I did not mean to click "Research integrity" issues and apologise for this.*

We would like to request the chairs to consider addressing this issue on open review as this research integrity flag has major consequences and cannot remain public as such.


We would also like to summarize our novelty and main contributions:

1) From a *conceptual* point of view, our work presents a new persistent-bias concept. We calculate the minimum value of $(x_i - x_j) - (x'_i - x'_j)$ in the Optimization Problem Objective Function (1). A positive value for the objective function means that *the input is more biased towards the correct class than its counterfactual, in the entire region*. In other words, a positive objective function means $(x_i - x_j) - (x'_i - x'_j)>0$, that is $(x_i - x_j) >(x'_i - x'_j)$. This means that $x_i$ relative to $x_j$ is larger than $x'_i$ relative to $x'_j$, which highlights the bias of the original input towards the correct class, compared to its counterfactual, in an entire region.

2) From a *methodological* point of view, we provide a **sound** formulation for identifying persistent bias, with **hard** formal guarantees over **an entire region**, as opposed to other techniques. Our formulation is hard (as opposed to other works that provide soft guarantees, e.g., Nanda et al.) and sound (when we prove a property, it is guaranteed to hold). As mentioned above, if our optimization problem has a positive solution (a solution that leads to positive objective function value), it essentially means that the input is more biased towards the correct class than its counterfactual, in the entire region (persistent bias). And, this is guaranteed to hold (**formally proved**), due to the formal foundation of our approach and its soundness property. In addition, our formulation allows peeling the softmax non-linearities exactly and without introducing any additional over-approximation, thanks to the relativity of the measure we define (please see the Equation in Lines 134-137). This leads to a linear objective function for the optimization, enabling the adoption of efficient linear programming solvers.

3) From an *experimental* point of view, we would like to reiterate that our approach identifies **formally-proved** instances of **persistent** bias that the other approaches do not, as Tables 1, 2, 3 show (The darker the gray shade is, the more instances of bias are identified). Considering Table 1 on the Adult dataset (gender, $\delta = 1$), CertiFair (100,50) shows almost no disparity in RB (95.6% vs. 96.5%) and only minor gaps in SP, EO, and CFA. Yet CPPBA uncovers a much larger confidence gap: 47.5% for Females versus 32.6% for Males.

**In summary, we present a new persistent-bias concept as well as a sound formulation for identifying persistent bias with coupled perturbations over an entire region, and we show experimentally that our approach identifies many instances of persistent bias, substantially more than the state of the art.**


Finally, we would like to highlight that:

* Reviewer XuoK had raised their score and were interested in continuing the discussion: *“I appreciate the authors discussing their paper and my review in more detail. In light, I have updated my review, and score, and would be more than happy to continue discussion.”*

* Reviewer GqzJ did not engage in the discussion. Unfortunately, Reviewers fiiY and oGie did not have sufficient time to respond after our clarifications were posted.

Thank you again for your time and commitment to maintaining a constructive review process.

---

### Meta-Review · Area_Chair_WtRd · 2025-12-23

**Summary:**

After reading the manuscript, reviewer comments, and authors response, I made my recommendation reject. Here are the detailed meta review.

**Research Question**

The authors consider the individual-level fairness.

**Motivation**

This paper lacks clear motivations. The authors argue that the existing studies on individual-level fairness suffer from scalability issues in high-dimension or complex architecture.

Unfortunately, the proposed technique proposes a new fairness notion, and did not tackle the above challenge. Therefore, my meta review is short. A clear reject is recommended.

**Reviewer Concerns:**

Reviewers have concerns as follows:

1. Motivation, the value of the proposed PBA.

2. Experiments. The experimental part is no strong. More datasets and large models are suggested.

3. Novelty. The idea of PBA seems incremental.

Although the authors provide the response, unfortunately none of them are well addressed.

**Reviewer Scores:**

I do not think the reviewers would change their score. Especially, this paper suffers from the non self-standing issue, that I mentioned above.

---

### Decision · Program_Chairs · 2026-01-26

Reject